# A quantitative geospatial analysis of the risk that Boko Haram will target a school

**Lirika Sola, Youdinghuan Chen, V. S. Subrahmanian**

Department of Computer Science and Buffett Institute for Global Affairs, Northwestern University, Evanston, IL, USA

* vss@northwestern.edu

**Editor:** Jessica Leight, IFPRI: International Food Policy Research Institute,Poverty, Gender and Inclusion,1201 Eye Street, UNITED STATES OF AMERICA, Washington, District of Columbia, 20005

**Data availability statement:** All relevant data are within the paper and its Supporting Information files.

## Abstract

We provide a novel quantitative geospatial analysis of school attacks perpetrated by Boko Haram in Nigeria. Such attacks are used by Boko Haram to kidnap boys (for potential use as child soldiers and suicide bombers) and girls (for potential use as domestic servants, as sex slaves, and suicide bombers). We first build a novel geospatially tagged data set spanning almost 15 years (July 2009 to April 2023) of data not only on Boko Haram attacks on schools (our dependent variable) but also a set of 15 independent variables (or features) about other attacks by Boko Haram, locations of security installations, as well as socioeconomic and geospatial characteristics of the regions around these schools. Second, we develop a univariate statistical analysis of this data, showing strong links between three broad factors affecting attacks on schools: *Security* presence in and around a school, the Boko Haram *Activity* in the area around a school, and the *Socioeconomic* characteristics of the region around a school. Third, we train several predictive machine learning models and assess their predictive efficacy. The results show that some of these models can accurately quantify the likelihood that a school will be at risk of a Boko Haram attack. In addition, they cast light on the features that are most important in making such predictions. We then analyze learned decision trees to identify some conditions on the independent variables that help predict Boko Haram attacks on school. Fourth, we use these decision trees to formulate multivariate hypotheses that we investigate further from a statistical perspective. We find that Security presence near schools, Activity of Boko Haram in regions, and the Socioeconomic factors characterizing the region a school is in are all significant predictors of attacks. We conclude with a policy recommendation.

## Introduction

On April 15 2014, the world woke up to the horrifying news that Boko Haram terrorists had kidnapped 276 schoolgirls from the town of Chibok. Almost 10 years later, Amnesty International reported that as of April 2023, 98 of these unfortunate girls were still being held by Boko Haram [1]. In another devastating attack in 2018, 110 schoolgirls were kidnapped in a Boko Haram attack on a school in Dapchi in Yobe State [2] though most were subsequently released. In December 2020, over 300 schoolboys were abducted from their school in Kankara

**Funding:** The author(s) received no specific funding for this work.

**Competing interests:** The authors have declared that no competing interests exist.

in Katsina state [3]. More recently, in March 2024, more than 300 schoolchildren were reportedly kidnapped by Boko Haram in the town of Kuriga in Kaduna State [4]. Just one day earlier, another group of children were reportedly abducted by Boko Haram [5].

With a name that is synonymous with "[Western] Education is forbidden", the group has carried out over 4300 attacks during the 14-year period (July 1 2009 to April 30 2023) that this study encompasses [6]. 76 of these attacks targeted schools. Boko Haram's attacks have had a devastating toll, resulting in over 39000 casualties and forcibly displacing over 2.4M people who once called the Lake Chad region home [7]. It is estimated that over 15 million people have been adversely affected by the insurgency and the resulting counter-terrorism operations [8]. All of this carnage has been caused by a group that, in 2022, was estimated by the US Office of the Director of National Intelligence (ODNI) to have only around a 1000 members [9].

Boko Haram's attacks on schools serve a multitude of purposes. On the one hand, opposition to Western education is a major plank of Boko Haram's *raison d'être*. Hence, attacks on schools can be viewed as a way to further this ideological goal. On the other hand, such attacks often go hand in hand with the kidnapping of the targeted school's students. Kidnapped students, boys, and girls, are often used as child soldiers, spies, and suicide bombers, frequently leading to horrific outcomes for the children involved. Girls are also subject to horrific abuse in the form of extreme domestic servitude and sexual slavery. Yet another possible reason for kidnappings is ransom. Even in those cases when some of the kidnapped children are released, the fate that awaits them is not always pleasant. Because of the stigma of association with Boko Haram, they are often shunned by the communities they grew up in, as well as others. Protecting the youngest members of society is an imperative for all nation-states.

The goal of this paper is to develop a data-driven, quantitative assessment of the risk of schools in Nigeria to an attack by Boko Haram. We will frequently refer to such attacks as "school attacks". We hope that our results, findings, and policy recommendation can be used by the Government of Nigeria to increase the security of schools.

Our model is data-driven, combining statistical and machine-learning methods. We do *not* claim to have invented new statistical or machine learning techniques in this paper. The contributions of this paper are our new dataset, the findings that predictively link Boko Haram's attacks on schools to the triad of Security Presence, (Boko Haram) Activity, and Socioeconomic factors, and the recommendation that we make to the Nigerian Government and the international community. This triad is based on the following hypotheses:

1. *Hypotheses 1 (Activity).* Boko Haram is more likely to target schools in regions where they are carrying out other operations in the region. This hypothesis is based on the Routine Activity Approach theory, suggesting that the proximity of schools to regions where Boko Haram is active renders them more likely targets. This increased risk is due to the offender's familiarity with the area, ease of access, and a potentially reduced risk of being intercepted [10]. Additionally, detailed studies of criminal geography and past findings [11] show that "juveniles do not commit property crimes in their immediate home areas to avoid recognition". At the same time, criminals do not go too far away from their home areas as they prefer to stay in areas they know well as shown by [12], who stated "Distances between home and crime-site were short". [13] showed that in the case of improvised explosive device (IED) attacks in Baghdad during the Iraq war, the locations of weapons caches facilitating those attacks were neither too near nor too far from the locations of those attacks. Based on these past findings, we use Boko

Haram's other types of attacks as a set of features that serve as a proxy for their location in this paper.

2. *Hypothesis 2 (Security Presence).* Boko Haram is less likely to target schools in regions with a heavy security presence. This hypothesis is driven by past work in criminology showing that the presence of police stations in locations with lots of crime is linked to subsequent reductions in crime in that location [14]. The hypothesis also aligns with the Deterrence Theory, which states that the visible presence of security acts serves as a deterrent to criminal activities, by increasing the perceived risk of detection and subsequent punishment for offenders [15,16]. A related study found similar results before and after the closures of police stations in the German state of Baden–Württemberg [17]. Likewise, a study of property crime in Buenos Aires [18] "found that the commission of crimes increases exponentially as the distance from the nearest police station increases". And even in Nigeria (albeit in the southwest of the country), [19] found that crime increased "as distance from police stations increased". Hence, we include features in our machine learning models that are related to security presence.

3. *Hypothesis 3 (Socioeconomic).* Boko Haram is more likely to target socioeconomically weaker areas than wealthier areas. This hypothesis is based on the theory of Social Disorganization, which suggests that areas burdened with socioeconomic factors such as unemployment, poverty, and inequality often struggle to maintain strong social controls and a united community. This leads to social disorganization, where the community cannot uphold standards that prevent behavior such as crime or terrorism [20,21]. Hence, our machine learning models include features related to the socioeconomic status of the region that a school is in.

4. *Hypothesis 4 (Geospatial).* Boko Haram is more likely to target rural areas than urban ones. This hypothesis is based on the idea that because rural regions are more geographically spread out, they are harder to defend than urban areas. A counter-argument could be that urban areas offer richer pickings for the criminals. This hypothesis is inspired by works such as [22] who investigated the relationship between urbanization and crime in regions including Rio de Janeiro, Karachi, and Lagos. This is the rationale underlying some of the features used by our machine learning models.

It is important that we are stating the above as hypotheses, not as statements of fact. To formally evaluate these hypotheses and to build machine learning models and quantitative risk scores, we have assembled a dataset spanning a period of almost 14 years (July 1 2009 to April 30 2023). The dataset combines information from a wide variety of open sources and captures the following *sub-datasets* related to (i) school attacks carried out by Boko Haram (our dependent variable), (ii) locations of schools in Nigeria, (iii) data on whether wards in Nigeria are rural or urban (a ward is a geographic administrative unit in Nigeria. The country is divided into 744 Local Government Areas or LGAs, each of which is further divided into 10–20 wards), (iv) data on other Boko Haram activity in Nigeria, (v) locations of security installations such as police stations and military bases in Nigeria, and (vi) socioeconomic characterizations of the wards in Nigeria. *The dataset has been made available as part of the Supplementary Materials.* We discuss our combined dataset in greater detail in the later sections of this report.

After the initial statistical evaluation of the hypotheses referenced above, we provide a machine-learning characterization of the risk of a Boko Haram attack on every school in Nigeria. We use cross-validation to identify the best such machine learning models. Because decision trees [23,24] generate predictive rules that are relatively easy to explain and understand, we use some of the decision trees learned from our study to identify additional hypotheses which are subsequently evaluated statistically.

We draw the following conclusions:

1. Ablation testing suggests that *Presence of Security Forces* is the single most important factor that mitigates attacks on schools by Boko Haram. Simply put, the more security installations there are near a school, the less likely it is to be attacked by Boko Haram. We therefore recommend that the Government of Nigeria and international donors focus on directing security budgets to creating new security installations in high-threat areas of Nigeria. Fig 1 shows our new country-wide risk map for the schools in Nigeria, while Fig 2 zeros in on a small part of Nigeria and shows a detailed risk map.

2. Unsurprisingly, *Other Boko Haram Activity* in a region is positively correlated with attacks on schools in that region.

3. Perhaps more surprisingly, wealth of a region is not clearly inversely correlated with school attacks. Two of our socioeconomic indicators show this, but a third shows the reverse trend. We discuss this seeming anomaly and provide some potential explanations of why this might be occurring.

4. Finally, we present 7 complex multivariate hypotheses about which schools are attacked, each learned from our decision trees, that are statistically validated.

Last but not least, we present comprehensive maps that indicate our predicted scores denoting the risk of school attacks by Boko Haram. Such risk maps will need to be updated

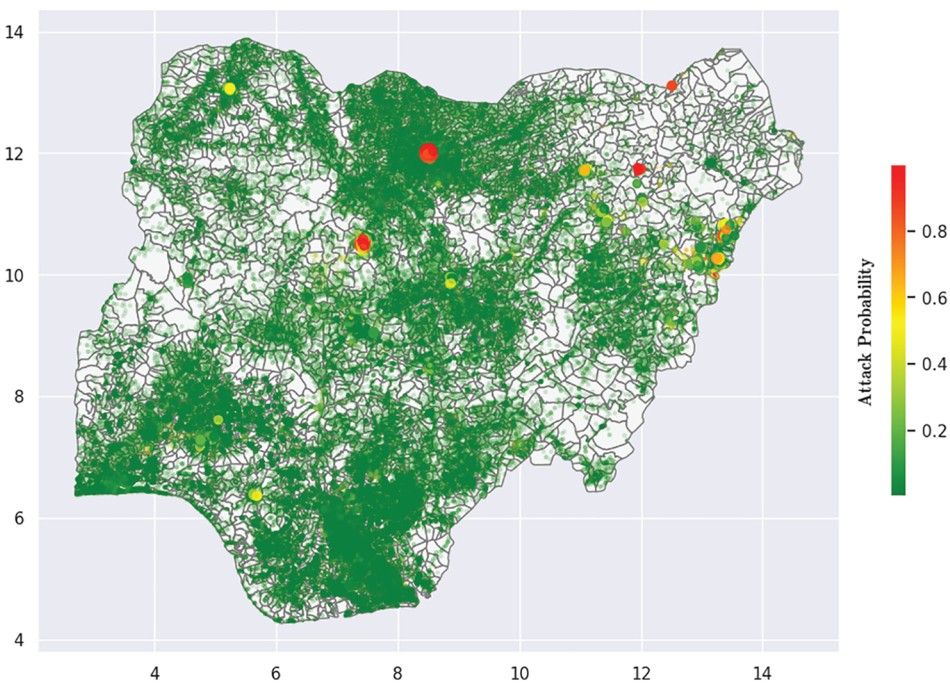

**Fig 1.   Predictive Risk of School Attacks by Boko Haram Across All of Nigeria.** These predictive risks are probabilities computed using the best performing model (namely AdaBoost, as presented later in the paper) to predict which school is most likely to be attacked. Maps were visualized using GRID3's Operational Wards data under the Creation Commons Attribution License (CCBY 4.0). https://data.grid3.org/datasets/GRID3::grid3-nga-operational-wards-v1-0/about.

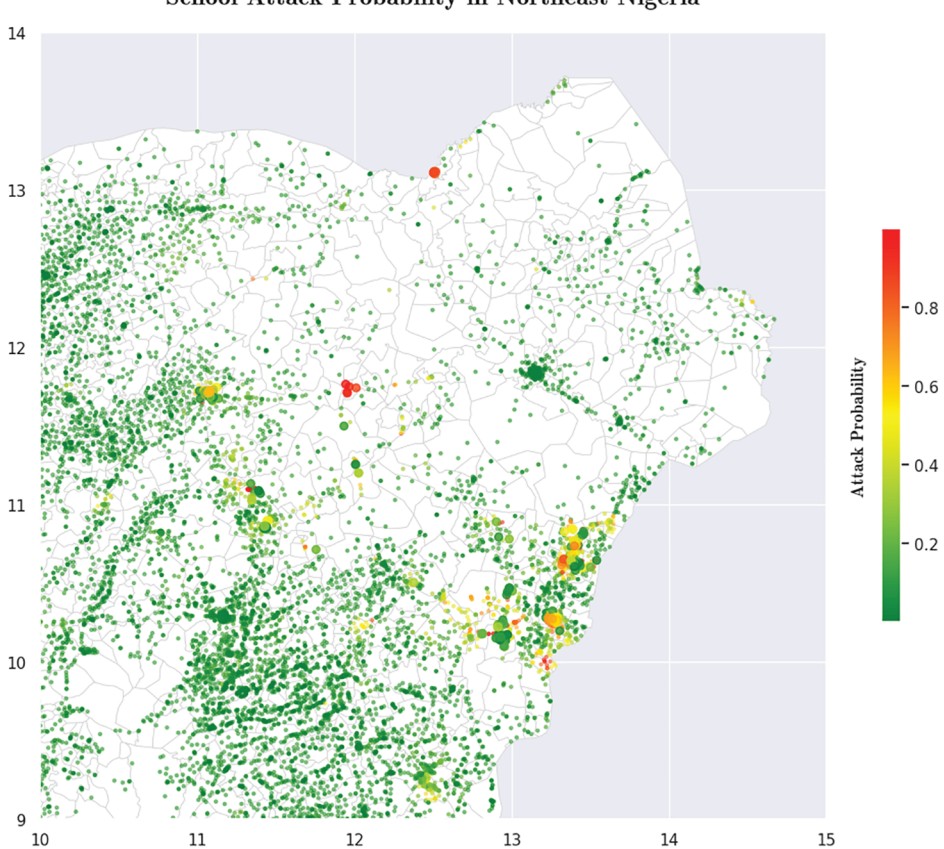

**Fig 2.   Predictive Risk of School Attacks by Boko Haram in Northeastern Nigeria.** These predictive risks are probabilities computed using the best performing model (namely AdaBoost, as presented later in the paper) to predict which school is most likely to be attacked. Maps were visualized using GRID3's Operational Wards data under the Creation Commons Attribution License (CCBY 4.0). https://data.grid3.org/datasets/GRID3::grid3-nga-operational-wards-v1-0/about.

over time as new Boko Haram activity, new security installations, and new socioeconomic conditions, change the security landscape in Nigeria.

## Boko Haram: A rief history

To set the context for this study, we provide a brief overview of the history of Boko Haram. Readers interested in a comprehensive history of Boko Haram can consult more detailed studies such as [25–28].

According to the International Monetary Fund (IMF) [29], Nigeria had a per-capita gross domestic product (GDP) of $ 1760 in 2023, placing the country's wealth in the upper quartile across African nations. However, this wealth is unequally distributed. The south of the country has greater wealth, in part because of the presence of oil in the Niger Delta [30]. This has led to a situation where the northern part of the nation has experienced greater food insecurity [31] and unemployment than the south. For instance, [32] states that "High unemployment has been blamed for civil unrest in Nigeria, in some cases leading to a revolution e.g. Boko Haram crisis in the Northern part of the country." Similarly, [33] states that "Structural

unemployment and widespread poverty are believed to be the basis for the activities of miscreants such as militant youth in the Niger Delta and the present deadly Boko Haram in northern Nigeria upsetting the seemingly peaceful and stable political situation."

A religious divide between the largely Muslim north and the largely Christian south of the country has also played a role in the conflict. Since the late 1990s, Sharia law has been introduced in several northern states, leading to " national and sub-regional terrorism with crucial ramifications for national development as well as national security" [34].

It was in this climate that Boko Haram was founded in 2002 in Maiduguri by a cleric, Mohammed Yusuf, in the Northeast of the country. During the next several years, Boko Haram's violent activities gradually increased and included attacks on security forces, attacks on Christian populations, and kidnappings. As recounted vividly in [35], the conflict between Boko Haram and the Nigerian state peaked in 2008-2009. Nigerian security forces carried out Operation Flush in 2008, which led to a spate of retaliatory attacks by Boko Haram in 2009, killing over 700 people. This was the beginning of what is called the Boko Haram Uprising. Nigerian forces responded vigorously, killing over 1000 people and executing Mohammed Yusuf, the group's founder, in custody. This was followed by a "decade of terror" [27]. During the 2009–2020 period, Boko Haram, under their new leader, Abubakar Shekau seized territory, and carried out dramatic attacks on numerous security facilities including both Nigeria's Police Headquarters and the UN Headquarters in Abuja in 2011. Over the next few years, churches, schools, colleges, police stations and army barracks were attacked viciously by Boko Haram. On April 15 2014, tax day in the US, the world woke up to the news that Boko Haram had kidnapped 276 schoolgirls from Chibok, leading to the ♯BringBackOurGirls campaign [36,37] that was supported by numerous celebrities including then U.S. First Lady Michelle Obama. Shortly thereafter, the organization used one of the unfortunate victims of the Chibok attack as a suicide bomber [38]. Boko Haram has repeatedly used girls as suicide bombers in subsequent years [39].

Despite a spate of horrific attacks, only a few of which are listed above, in December 2015, the Nigerian government claimed that Boko Haram was defeated [40]. A similar claim was repeated by Nigeria's Information Minister in October 2019 [41]. Such claims were repeated several times in subsequent years, despite evidence that Boko Haram has consistently managed to carry out attacks. To this day, Boko Haram continues to carry out attacks - a phenomenon supported by attacks such as one in Maiduguri in November 2023 that reportedly killed 40 people [42], as well as a January 2024 attack that reportedly killed 14 people in Yobe state [43].

## Child kidnappings

Child kidnapping has been used as a strategy by Boko Haram since the beginning of the insurgency in 2009. Kidnapped girls are forced into a combination of domestic and sexual servitude [44]. They are also used as spies, fighters [45] and suicide bombers [46]. Children of both sexes are often starved, physically abused and forced to torture and kill civilians [47]. In addition to the horrific attacks carried out by the victims of Boko Haram's abductions, it is clear that victims of kidnappings face an extreme form of child abuse with profound mental health and other consequences [48,49]. To make matters worse, efforts to reintegrate Boko Haram's male fighters (including boys who were abducted and trained to become fights) into society through programs such as Operation Safe Corridors [8,50] have not been very successful and in fact, as stated by [8], Nigerian communities often do not want take back and reintegrate fighters and women and children who were themselves victims of Boko Haram [51].

## Materials and methods

We now describe: (i) how we created our Boko Haram dataset, (ii) our statistical analysis of the univariate hypotheses posed in the Introduction, (iii) our machine learning analysis of our Boko Haram data, and (iv) our statistical analysis of several multi-variate hypotheses inspired by our machine learning analysis in (iii) above. Finally, (iv), we show the risk maps we have generated for Nigerian schools.

### Our data

In order to explore a set of statistically testable hypotheses and to build out machine learning models underlying our quantitative risk scores, we created a dataset spanning a period of almost 14 years (July 1 2009 to April 30 2023). The dataset combines information from a wide variety of open sources, as noted in Table 1, and captures the following "sub" datasets relating to school attacks carried out by Boko Haram (our dependent variables), locations of schools in Nigeria, (iii) data on whether wards in Nigeria are rural or urban, (iv) data on other Boko Haram activity in the vicinity of schools, (v) locations of security installations in Nigeria, and (vi) socioeconomic characterizations of the wards in Nigeria. Our use of reliable sources such as ACLED [6] (for the dependent variable), GRID3 for school data, police stations and socioeconomic risk factors [52], and OpenStreetMaps [53] for military installations also eliminated the need for imputation methods. We will discuss our combined dataset in further detail later in the paper.

### Boko Haram's Attacks, 2009–2023

Fig 3a shows the total number of attacks carried out by Boko Haram in our dataset, as well as the total number of attacks on schools (Fig 3b). We note that there are challenges in determining whether an attack should be attributed to Boko Haram or not. We relied on the widely used and highly respected ACLED data set [6] for this purpose. ACLED data showed over 4300 Boko Haram attacks in all during the 2009–2023 period, of which 76 targeted schools. The bulk of these attacks occurred during the height of the Boko Haram insurgency (2012–2014), but the numbers since then still indicate several attacks per year. Although ACLED records no attacks targeting schools in Nigeria during 2016 and 2019 (as shown in Fig 3b), Boko Haram remained active in the country during these years, as evidenced by Fig 3a. Though Boko Haram carries out attacks and has a presence in the nations of Cameroon, Chad, Niger, and Nigeria, the overwhelming majority of school attacks are in Nigeria. We therefore limited our study to school attacks in Nigeria.

We also investigated the months when Boko Haram is most likely to carry out an attack and/or a school attack. This information is shown in Fig 4. Fig 4a looks at non-school attacks, while Fig 4b shows school attacks. We see an interesting difference here. While non-school

**Table 1. Summary of data sources and key attributes.**

| Data | Source | Record number | Categories |
|---|---|---|---|
| Dependent variable | ACLED | 76 | School attacks |
| B.H activity | ACLED | 4224 | Arson, looting, kidnapping etc. |
| School data | GRID3 | 103,064 | primary, mixed, others, secondary, tertiary, pre-primary |
| Security installations | OSM & GRID3 | 920 | Police & Military |
| Risk factors | GRID3 | 36 | Communications, Exposure, Socioeconomic |

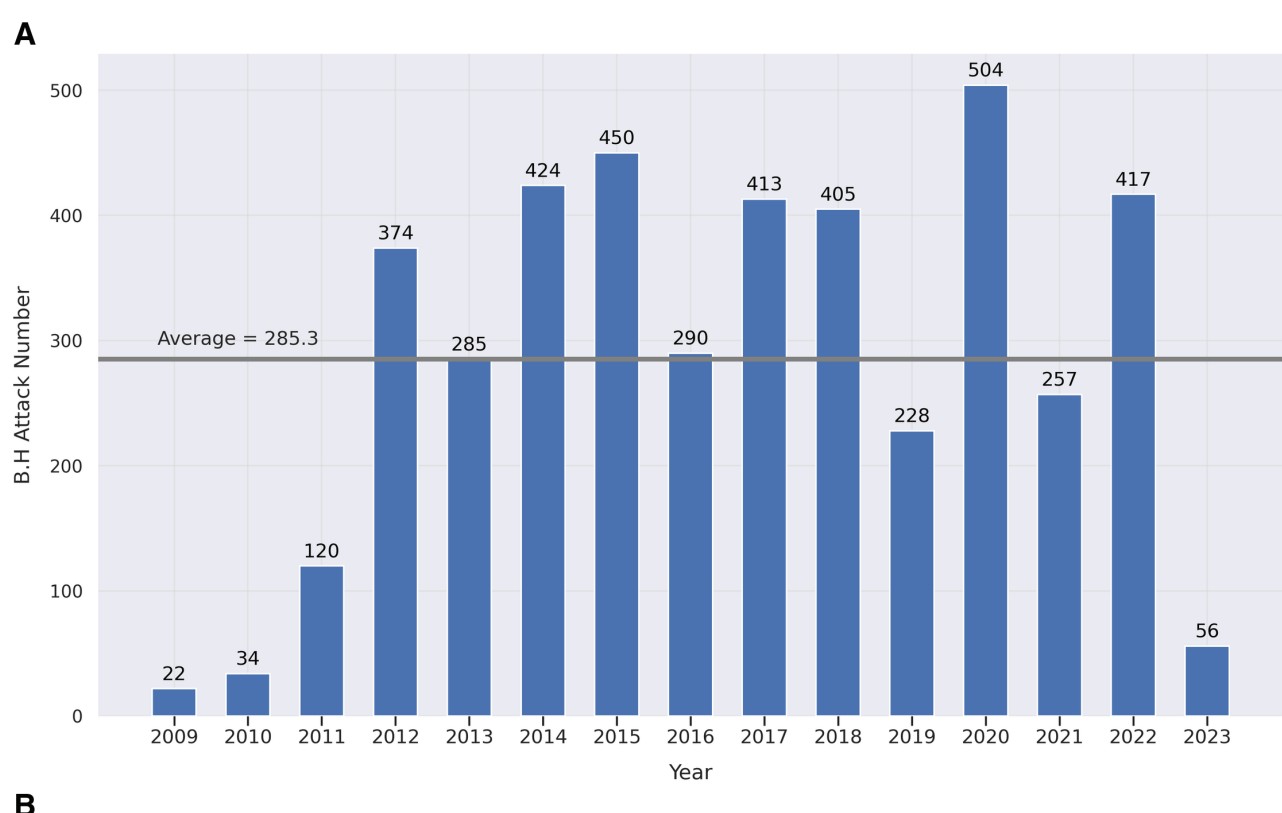

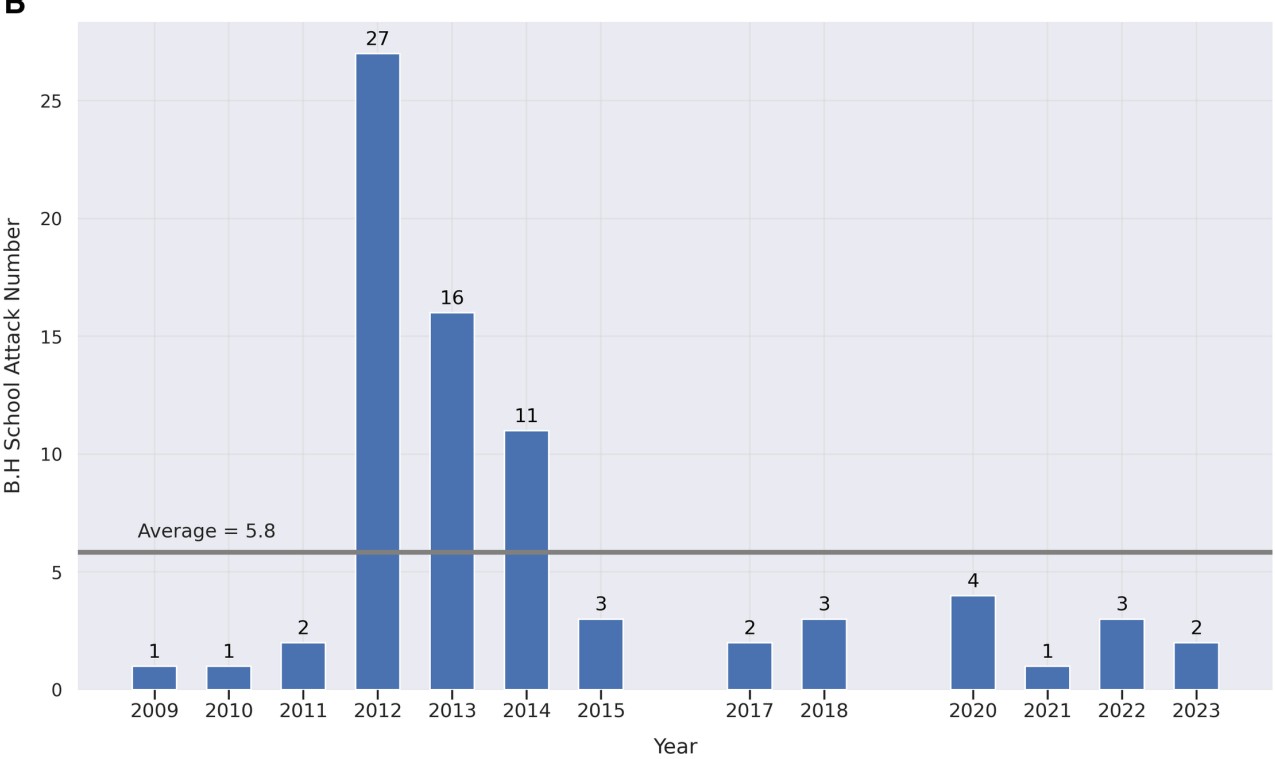

**Fig 3. Attacks by Boko Haram, July 2009–April 2023.** (a) shows attacks not including school attacks. (b) shows attacks on schools. Numbers only run through the end of April 2023. These charts are based on ACLED data [6].

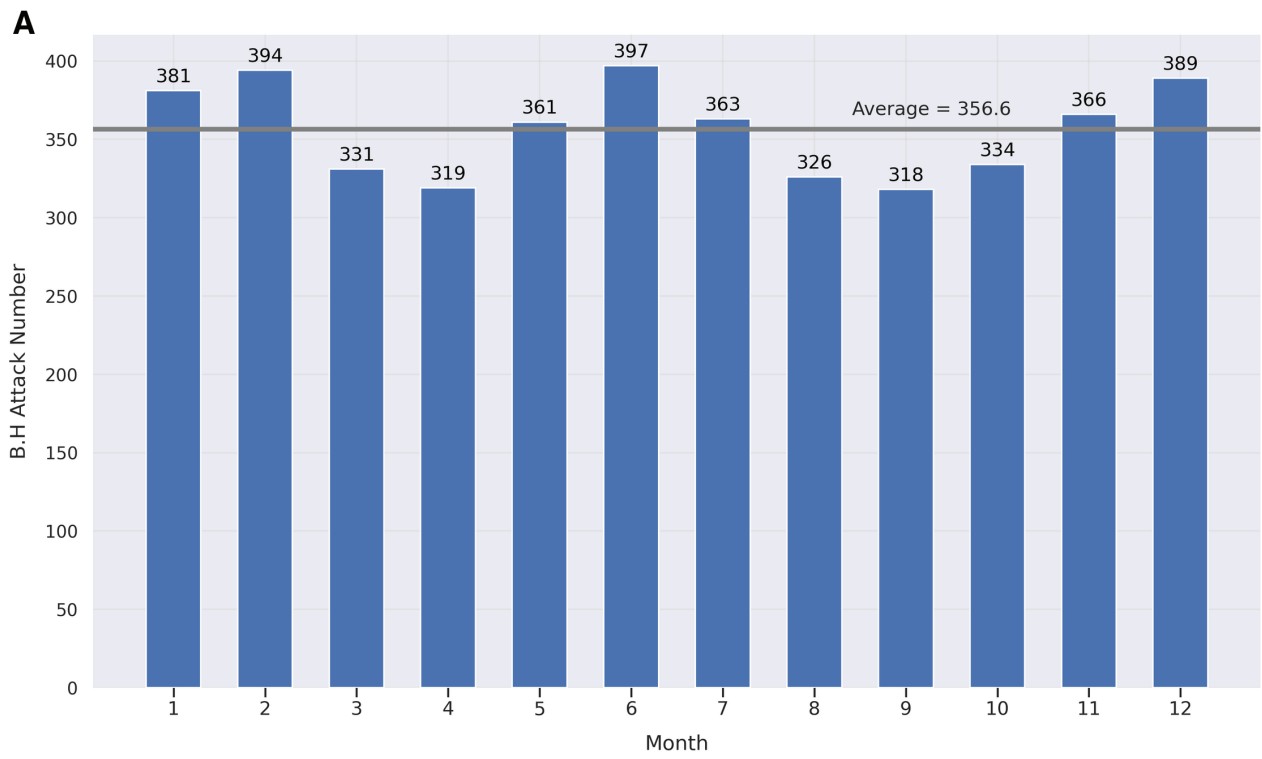

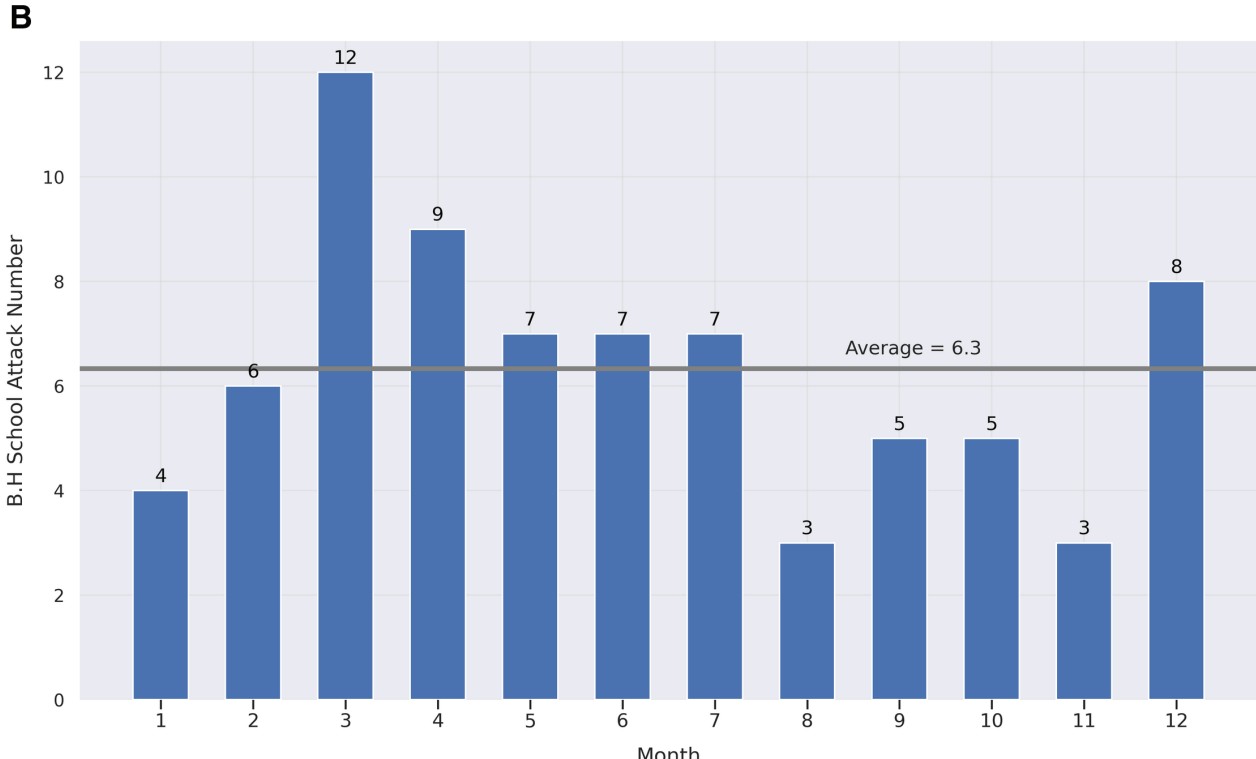

**Fig 4. Month-by-Month Attacks by Boko Haram, July 2009–April 2023.** (a) shows attacks not including school attacks. (b) shows attacks on schools. 2023 numbers only run through the end of April 2023. These charts are based on ACLED data [6]. Maps were visualized using GRID3's Operational Wards data under the Creation Commons Attribution License (CCBY 4.0). https://data.grid3.org/datasets/GRID3::grid3-nga-operational-wards-v1-0/about.

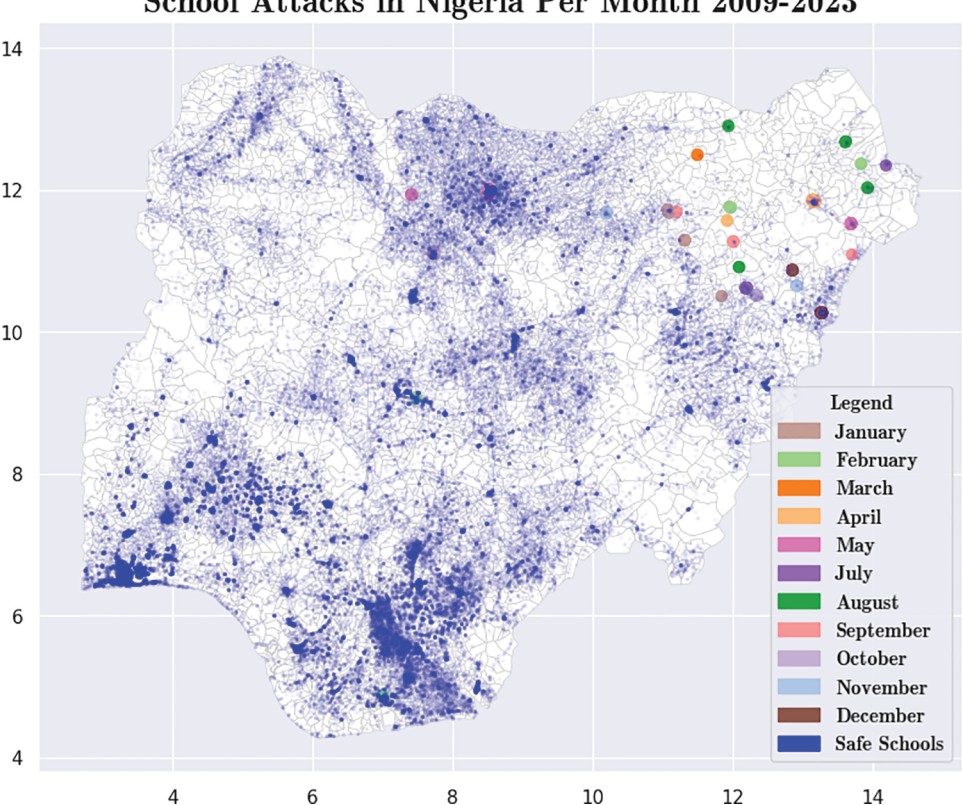

**Fig 5. Month-by-Month Locations of School Attacks by Boko Haram, July 2009–April 2023.** Blue dots show locations of schools that were never attacked. Dots with other colors show the month when a school was attacked. These charts are based on ACLED data [6]. Maps were visualized using GRID3's Operational Wards data under the Creation Commons Attribution License (CCBY 4.0). https://data.grid3.org/datasets/GRID3::grid3-nga-operational-wards-v1-0/about.

attacks are more or less uniformly distributed over the 12 months of the year, school attacks occur less frequently in January, August, and November. While school holidays may vary from one institution to another, the academic year in Nigeria typically runs from September to July. This accounts for the lower frequency of school attacks observed in August. Moreover, the school year is segmented into three terms, with most schools having midterm breaks and/or a winter break scheduled from mid-December to mid-January.

Fig 5 shows the locations of school attacks in Nigeria. The map shows that most attacks happen in Northeastern Nigeria in the Lake Chad region with a smaller number of attacks in other northern states.

## School data

Our dataset includes information about 103,064 schools in Nigeria including both primary and secondary schools, but not including universities and colleges. The data was obtained from the GRID3 Data Hub [52]. Each school, serving as the unit of analysis, has a unique ID, a latitude and longitude showing the location of the school, and the ward in which the school is located. In addition, each school has the following fields:

- *Education type* consisting of 4 possible types: formal, religious, informal, and integrated.

- *Management type* consisting of 13 possible types: public, private, faith-based, private faith-based, public NGO funded, schools funded based on a public-private partnership, public faith based, NGO funded, unknown, private NGO funded, faith-based NGO-funded, state government-funded, and federal government-funded.
- *Subtypes* consisting of 12 possible types: standard, primary, pre-primary, nursery, aggregate, adult education, mixed, others, tertiary, senior, junior, and university. In our study, adult and university-education instances were excluded as these do not involve schools.
- *Category types* with 6 possible values: primary, mixed, others, secondary, tertiary, pre-primary.
- *Type of education imparted* included formal vs. informal vs. religious education.
- *Type of management of the school* that included public school vs. private school.
- *Category to which the school belongs* consisting of (primary vs. secondary vs. "mixed" which includes both.

## Security installations

We obtained information on the location of security installations in Nigeria using two methods.

First, we continued the use of the GRID3 Data Hub to extract the locations of 802 police stations in the country [54]. For each police station, we captured an ID for the station, a latitude and longitude describing the location of the station, as well as the name of the state and the specific region in the state where the police station is located. It is important to note that the dataset is may be incomplete in terms of covering all police stations in the country. To enrich our analysis and achieve a more comprehensive understanding of security installations, we incorporated data on Nigerian military sites, including military installations, checkpoints, barracks, airfields, and training areas, obtained via the Overpass API [55]. OverPass is a software service that is built on top of Google Street Maps. It can be queried to obtain diverse forms of information. From this, we extracted locations of 128 military facilities. After checking for duplicate records, we obtained data on 920 security locations throughout the country. Fig 6 serves as a visual representation of the security installations and where they are located. The queries used to extract the military installations via the Overpass API can be found in S6 Appendix F.

## Socioeconomic Data

We gathered proxies for socioeconomic data about each school ward from the geospatial maps provided by GRID3 Data Hub [52]. In particular, we obtained three types of "risk scores", which assess the percentage of individuals in households within the wards who face socioeconomic disadvantages. The scores range from 1 to 5 with 1 indicating the lowest risk and 5 the highest. The score scale is determined by taking into consideration and consolidating multiple data sources including household and population surveys as well as other agencies such as USAID, United Nations, World Bank, USGS, WorldPop and more. The predefined scale of risk categories at the ward level was also omitted the potential presence of outliers in the data, thereby eliminating the need for outlier handling. The score categories are defined as follows:

- *Communication Risk Score [56]:* This metric is based on the level of communications access that the population in the ward has. It is based on the percentage of members of the ward with access to radios, TVs, the Internet, and other sources of news. Fig 7 presents a color-coded map that illustrates the communication risk scores at the ward level across Nigeria.

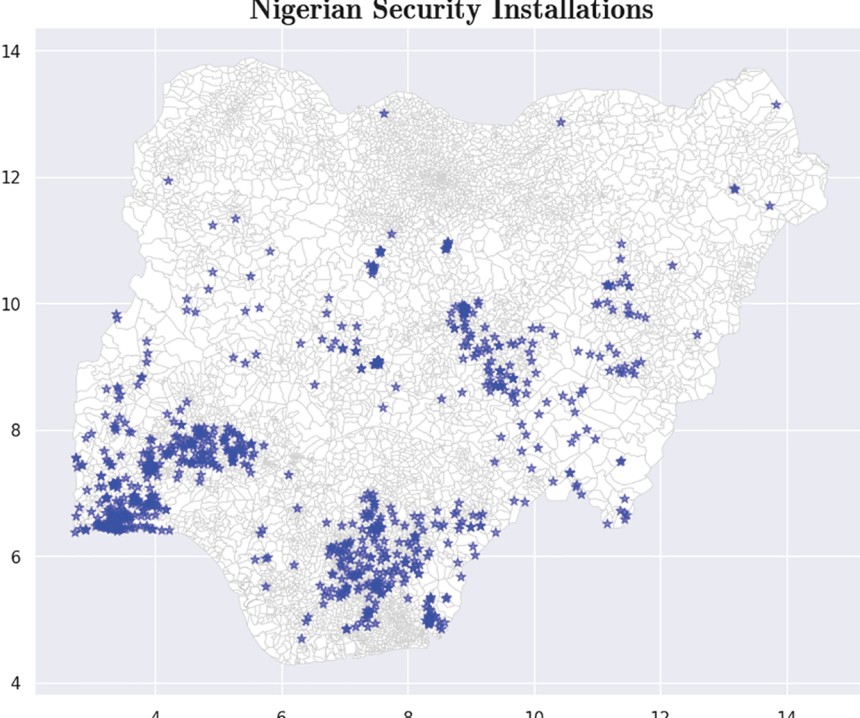

**Fig 6. Location of Nigerian security installations.** The blue stars represent security installations, the majority of which are situated in the southwestern states. These charts are based on Open Street Maps [53] & GRID3 data [52]. Maps were visualized using GRID3's Wards data under the Creation Commons Attribution License (CCBY 4.0). https://data.grid3.org/datasets/GRID3::grid3-nga-operational-wards-v1-0/about.

- *Exposure Risk Score [57]:* This metric is an aggregate that captures indicators such as population density, proximity between households, water, sanitation, and hygiene. Fig 8 presents a color-coded map that illustrates the exposure risk scores at the ward level across Nigeria.
- *Socioeconomic Risk Score [58]:* This metric captures the overall level of socioeconomic risk. Fig 9 presents a color-coded map that illustrates the socioeconomic risk scores at the ward level across Nigeria.

We can think of these "risk scores" as proxies for wealth. Small scores (close to 1) typically relate to areas that are more wealthy than those with higher risk scores.

## Other derivative data

For each attack location, we found the 5 nearest security installations using the Shapely and GeoPandas tools' Ball-tree approach [59] together with the Haversine distance metric [60] which is specialized for use with latitude-longitude data. The same method was used to find locations of the 5 nearest attacks to a school.

S1 Appendix A in the Supplementary Material contains a list of all the independent variables (i.e. features) we associated with each school.

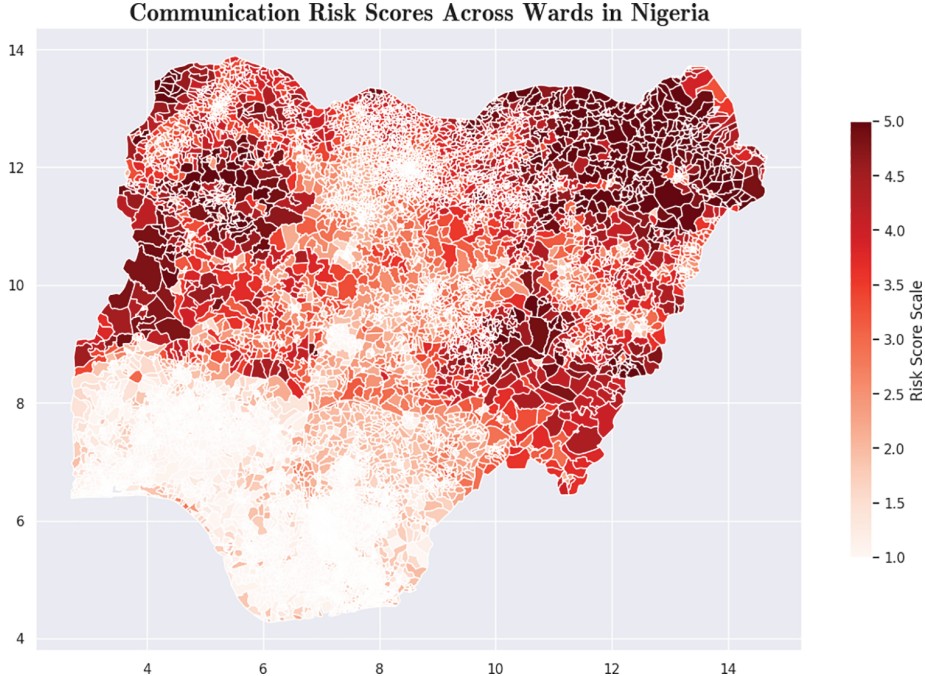

**Fig 7. Color-coded map of communication risk scores across Nigerian wards.** Maps were visualized using GRID3's Operational Wards data under the Creation Commons Attribution License (CCBY 4.0). https://data.grid3.org/datasets/GRID3::grid3-nga-operational-wards-v1-0/about.

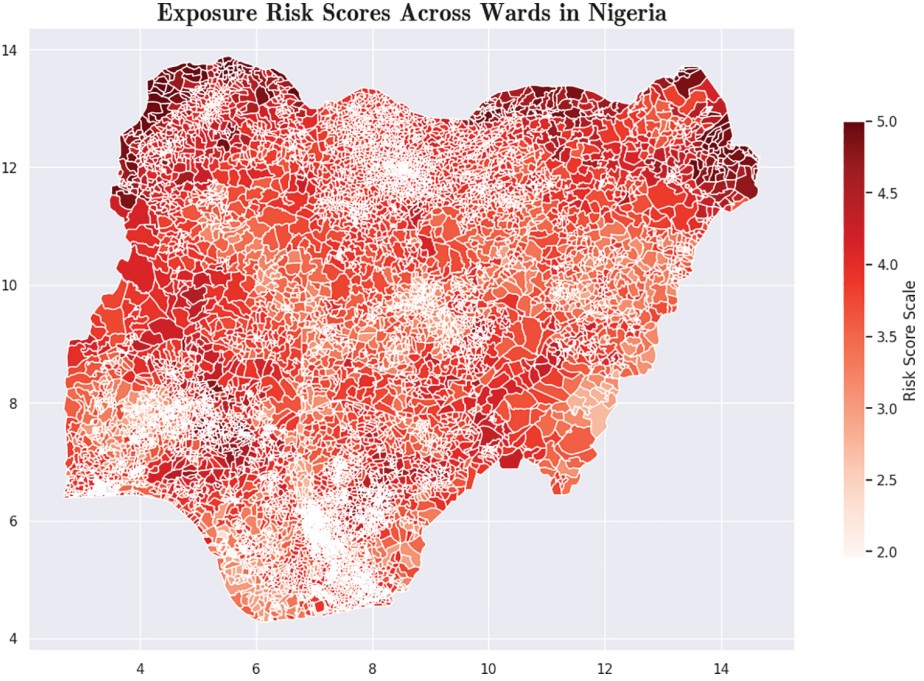

**Fig 8. Color-coded map of exposure risk scores across Nigerian wards.** Maps were visualized using GRID3's Operational Wards data under the Creation Commons Attribution License (CCBY 4.0). https://data.grid3.org/datasets/GRID3::grid3-nga-operational-wards-v1-0/about.

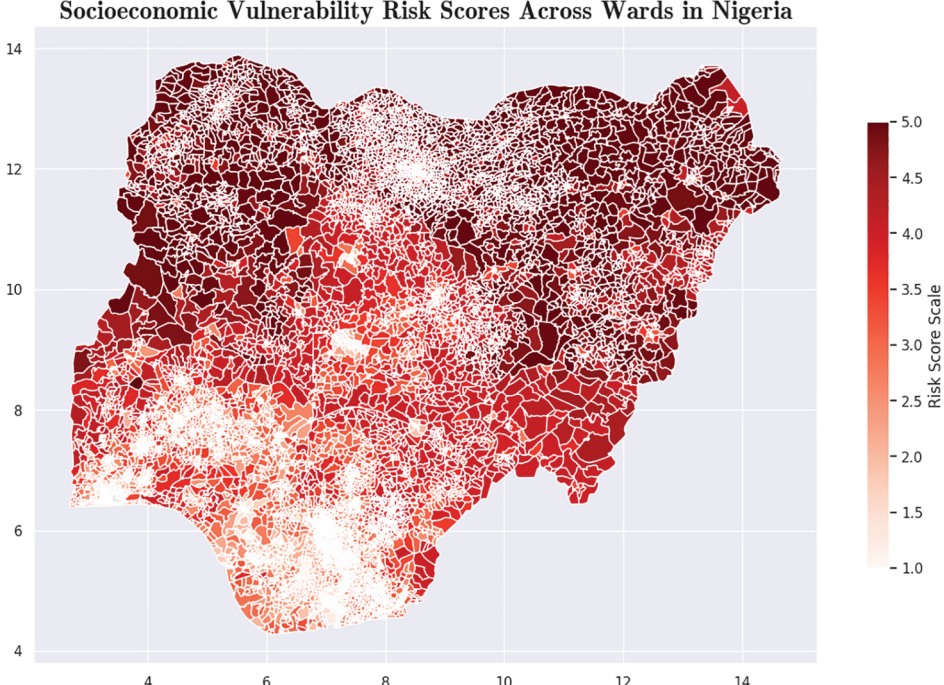

**Fig 9. Color-coded map of socioeconomic risk scores across Nigerian wards.** Maps were visualized using GRID3's Operational Wards data under the Creation Commons Attribution License (CCBY 4.0). https://data.grid3.org/datasets/GRID3::grid-nga-operational-wards-v1-0/about.

### Statistical inference of school attacks in relation to social attributes

In this section, we present the results for our first round of statistical analysis. We examine the relationships between school attacks and several variables (e.g. socioeconomic risk factors and wealth described in Hypotheses 1–4) below. Given a school, we examine whether a school attack happened within $k \in \{1, 2, 3, 5, 10\}$ km. of the school. We wanted to investigate how predictions of whether a school is at risk of attack vary in quality as we expand the window of what someone might consider threatening to the school. We investigated how predictive performance changes when the threshold is set to 1, 2, 3, 5, 10 km ranges.

There are several papers (e.g. [13]) that measure spatial prediction error by looking at the distance between the location where an event is predicted to occur, and the true location. The choice of $k$ captures this in a binary setting. When our algorithms say that a school $S$ will be attacked, will some school within the circle of radius $k$ km. be attacked? If so, we say the prediction is correct. This is similar to the notion of distance-based accuracy [61] in predicting locations of individuals or events in social media.

This framing has a natural interpretation from a defense perspective. Imagine being a parent whose child goes to school $S$. If a school within 2 km. of school $S$ was attacked, would you consider your child's school $S$ to be safe? Probably not. What about if the nearest school attacked was 5 km. away from your child's school? Would that make you feel safer? From a defense perspective, the rationale for studying proximate attacks at different distances is to account for the possibility of target selection based on characteristics we do not know about. Perhaps school $S'$ is 2 km. away from school $S$ and the only reason $S'$ was selected for an

attack was that the leader of the attacking party that day knew the neighborhood around $S'$ slightly better than the neighborhood around $S$. Or perhaps the group of attackers was closer to $S'$ on that specific day. As we cannot systematically gather such data, we consider a school $S$ to be attacked if any school within a distance $k$ of $S$ was attacked for $k \in \{1, 2, 3, 5, 10\}$ km, regardless of how many other schools lie within a distance $k$ from $S$.

At the 1 km threshold, our experiments will demonstrate a lack of ability to accurately predict if a school within 1 km of a given school S will be attacked. But for the 2, 3, 5, and 10 km levels, our experiments show that our best predictive model performs well.

Fig 10 represents a visualization of assigning the dependent variable to Nigerian schools when $k = 5$ km. These are the dependent variables we study in this paper.

**Hypothesis 1.** *For a given school, a given $k \in \{1, 2, 3, 5, 10\}$, we define "Class 1" to be the set of all schools s which experienced an attack within a distance of k kilometers (km) from s. "Class 0" is the set of all schools that did not experience an attack within k km of the school, i.e. the complement set of Class 1. We hypothesize that the total number of attacks (not just school attacks) by Boko Haram in Class 1 differs significantly from Class 0.*

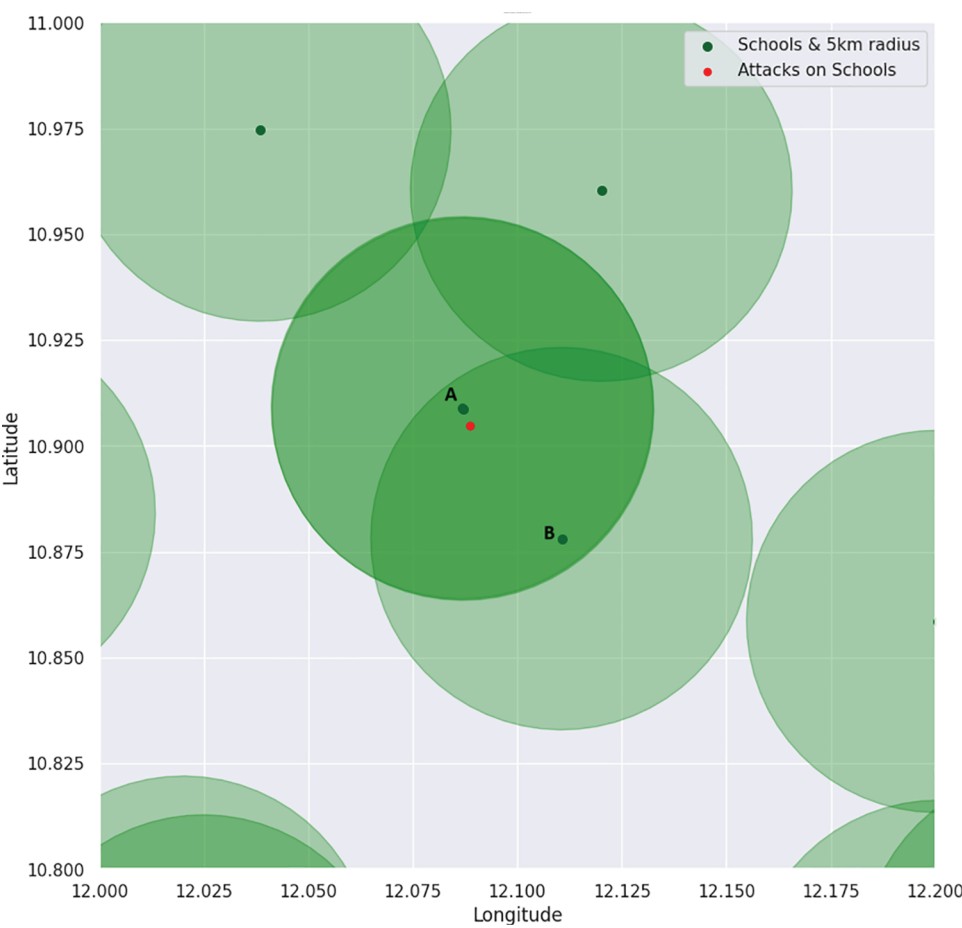

**Fig 10. Schools labeled A and B were identified as having experienced attacks by Boko Haram, due to the incidents occurring within a 5 km radius of their premises.**

Recall that Class 1 is the set of schools $s$ that experienced an attack nearby, i.e. within $k \in \{1, 2, 3, 5, 10\}$ km, the quantity we aim to predict. The geospatial granularity of the precision is more fine-grained when $k$ is small. The independent variables we examined are the total number of attacks within a distance of $d \in \{5, 10, 25, 50\}$ km. of the school which serves as a proxy for the presence of Boko Haram in the region near the school.

We formally compare the total number of attacks between the two classes using a linear model-based approach (two-sided Welch's $t$-test) between classes [62]. We report the mean difference along its 99% confidence interval (CI) and $P$-value between the number of attacks between the two classes, summarized in a forest plot. We then repeat the analysis for all $d \in \{5, 10, 25, 50\}$ km. When the difference is statistically significant (i.e. $P < 0.05$), the 99% CI does not contain the null value of 0 (i.e. no difference). To address multiple hypothesis testing but account for the correlation structure of the shared radii across the analyses, we corrected each raw $P$-value using the false discovery rate (FDR) method [63]. S10 Appendix J provides a detailed description of the statistical methods used in this paper.

*Results and Analysis of Hypothesis 1*. Fig 11 shows the results (Forest Plots, FDR-corrected $P$–values) of our analysis for each of the dependent variables based on $k$. Based on Fig 11, we can conclude that:

- There is a significantly higher number of attacks in Class 1 across all values of $d$ and all the 5 dependent variables (i.e. values of $k$).
- There is consistency in direction, i.e. the difference between the means of Class 1 and Class 0 increases as $d$ *and* $k$ increase.
- All results are statistically significant as $FDR \ll 0.05$ in all cases.

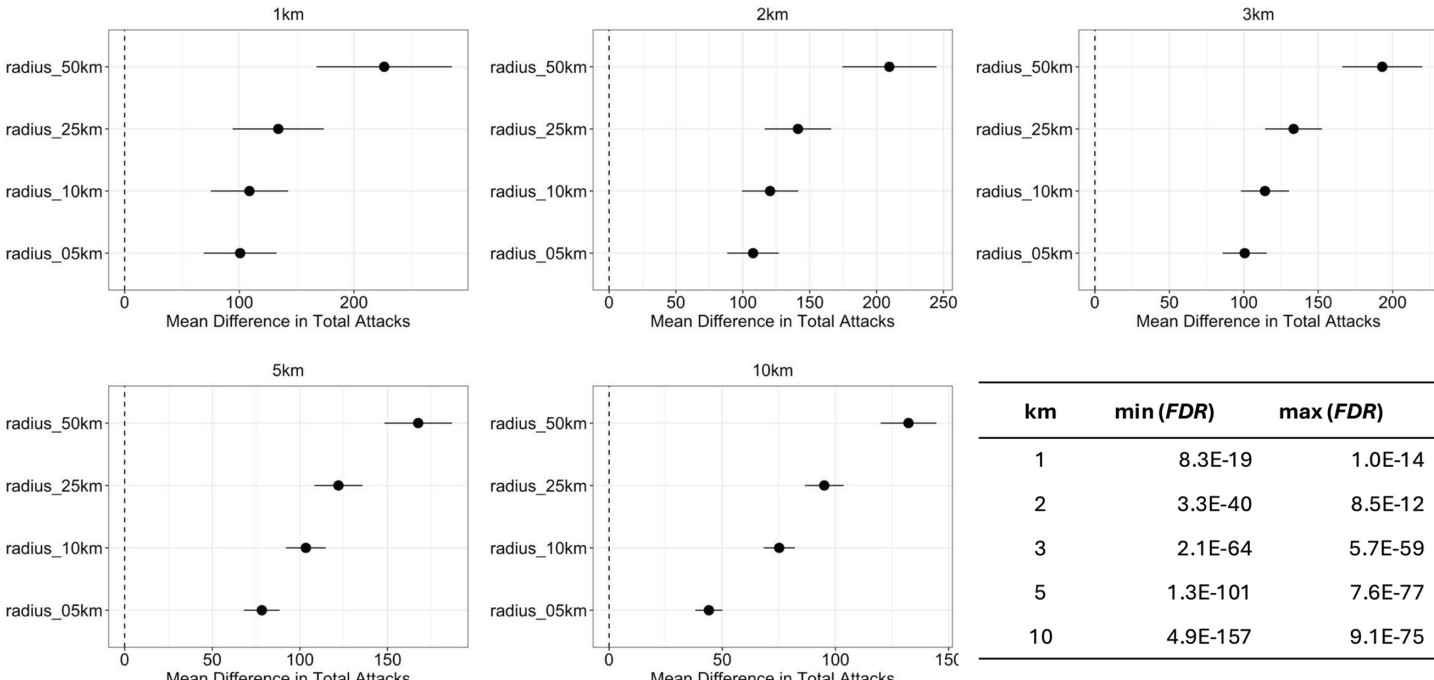

**Fig 11.   Hypothesis 1: Difference between means of "Class 1" (a school attack happens within $k$ km of a given school) vs. "Class 0" (no school attack happens within $k$ km of a given school) for $k \in \{1, 2, 3, 5, 10\}$ when considering the total number of other attacks that occur within $d \in \{5, 10, 25, 50\}$ of the school as the independent variables.** The exact numerical values of the point estimates and 99% CI can be found in Table S4, S5 Appendix E.

- In all values of $k$, there is a step-wise, upward trend in mean difference as $d$ increases supporting greater certainty in the total number of attacks.
- As $k$ increases (e.g. 1–3 km), the total number of attacks tend to decrease.

**Finding 1.** *In short, we can conclude that the Activity hypothesis is correct: the more Boko Haram activity (other attacks) occurs in a region, the more likely that school attacks will occur in that region.*

Our second hypothesis investigates whether proximity to security installations (e.g. police stations, military bases) is linked to whether a school was attacked. To test this, we investigated the distance between each school $s \in S$ and the distance to the 1st, 2nd,…, 5th closest security installation. We asked the question: Between the outcome Class 1 (a school attack happens within $k$ kms. of a given school) and Class 0 (no school attack happens within $k$ kms. of a given school), is there a statistically significant difference in the distance to the 1st, 2nd, …, 5th closest security measure?

**Hypothesis 2.** *Consider the difference between means of Class 1 (a school attack happens within k kms. of a given school) and Class 0 (no school attack happens within k kms. of a given school) for $k \in \{1, 2, 3, 5, 10\}$ when considering the distance between each school $s \in S$ and the distances to the 1st, 2nd,…, 5th closest security installation. There is a statistically significant difference in the distance to the 1st, 2nd, …, 5th closest security installation in Classes 1 and 0.*

We use the same statistical approach used in Hypothesis 1, and repeat the analysis for all km variations. Unlike Hypothesis 1, we address multiple hypothesis testing using the Bonferroni correction instead of the FDR method because the distances from the school to the 1st, 2nd, ... 5th closest security installations are conditionally independent. Fig 12 below shows the forest plots and p-values (raw and Bonferroni) for each dependent variable $k \in \{1, 2, 3, 5, 10\}$ km.

*Results and Analysis of Hypothesis 2.* Fig 12 shows the results (Forest Plots, raw and Bonferroni-corrected P-values) of our analysis for each of the dependent variables based on $k \in \{1, 2, 3, 5, 10\}$ kms. We can conclude from this figure that:

1. In all km variations $k \in \{1, 2, 3, 5, 10\}$ kms of the dependent variable, Class 1 (Attack) is significantly farther away from a security installation than Class 0 (No Attack).
2. The magnitude of the difference for the 1st,…, 5th closest security installations is always increasing. One point to note is that the 1st and 2nd closest security installations in the cases when $k \in \{5, 10\}$ km were very close.
3. At higher km. variations, the mean-difference estimates of the 1st, …, 5th closest become more similar to one other and more distinguishable.

**Finding 2.** *We therefore conclude with high statistical significance that the closer a school is to security installations, the less likely it is to be attacked.*

Our third hypothesis is whether Boko Haram would be more likely to target socioeconomically weaker areas than wealthier areas. This hypothesis is based on the theory of Social Disorganization, which suggests that areas burdened with socioeconomic factors such as unemployment, poverty, and inequality often struggle to maintain strong social controls and a united community. This leads to social disorganization, where the community cannot uphold standards that prevent behavior such as crime or terrorism [20,21].

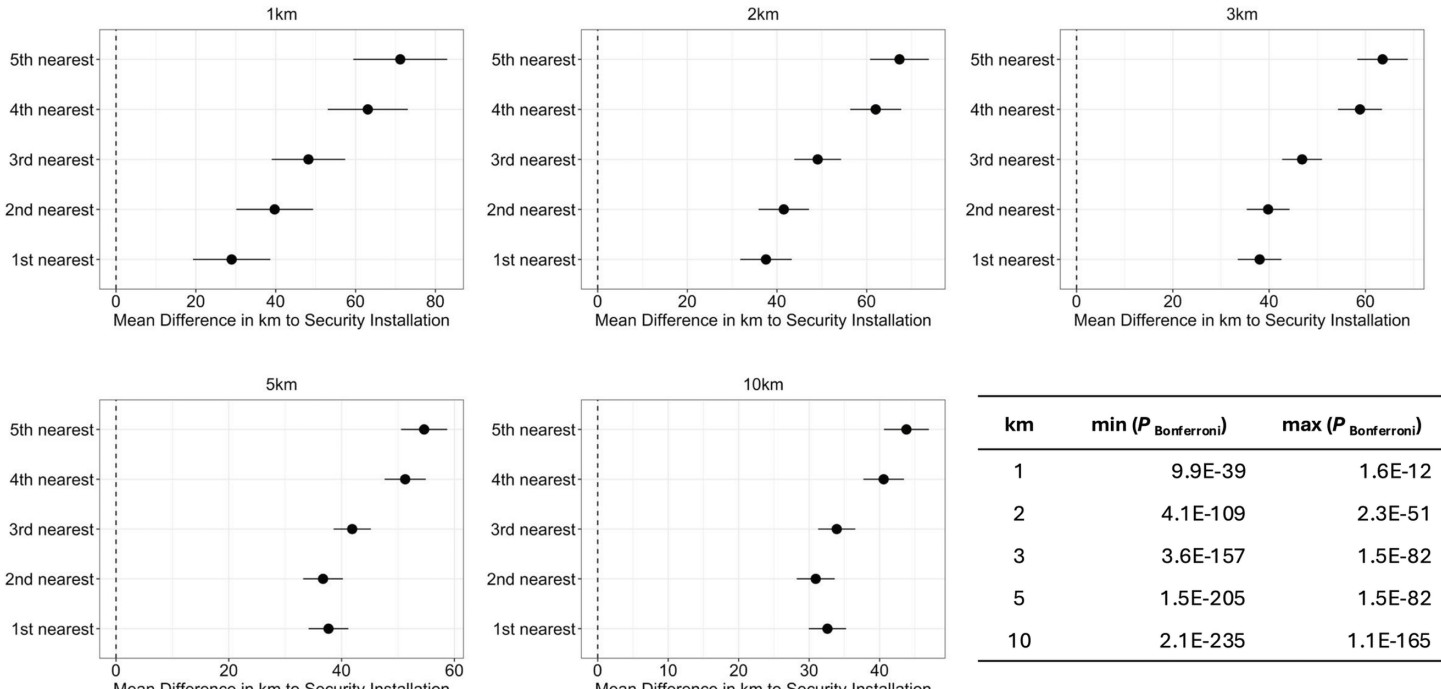

**Fig 12. Hypothesis 2: Difference between means of "Class 1" (a school attack happens within $k$ kms. of a given school) vs. "Class 0" (no school attack happens within $k$ kms. of a given school) for $k \in \{1, 2, 3, 5, 10\}$ when considering the distance between each school $s \in S$ and the distance to the 1st, 2nd, ..., 5th closest security installation.**

**Hypothesis 3.** *The difference between means of "Class 1" (now defined as a school attack happens within $k$ kms. of a given school) vs. "Class 0" (no school attack happens within $k$ kms. of a given school) for $k \in \{1, 2, 3, 5, 10\}$ increases as the Socioeconomic Risk Score of the region of each school decreases (i.e. the school is in a richer neighborhood).*

We used the same statistical method including the Bonferroni correction of raw *P*-values as in Hypothesis 2. Fig 13 shows the forest plots that summarize the finding.

*Results and Analysis of Hypothesis 3.* We were surprised to find a positive relationship between school attacks and the Exposure Risk, but a negative relationship between school attacks and each Socioeconomic Risk and Communication Risk (Fig 13). Specifically, we found that:

**Finding 3.**   1. *The magnitude of differences is consistent in the direction across all km variations of the dependent variable.*
2. *There is a negative association between school attacks and socioeconomic Risk Score and with Communication Risk Score.*
3. *There is a positive association between school attacks and Exposure Risk Score.*
4. *These results held for all $k \in \{2, 3, 5, 10\}$ km settings—but not for the 1 km variation. In the $k \in \{2, 3, 5, 10\}$ km settings, all risk scores individually showed highly significant differences between Class 1 and 0. The results remains statistically significant after the Bonferroni P-value correction, the most conservative approach for addressing multiple hypothesis testing.*

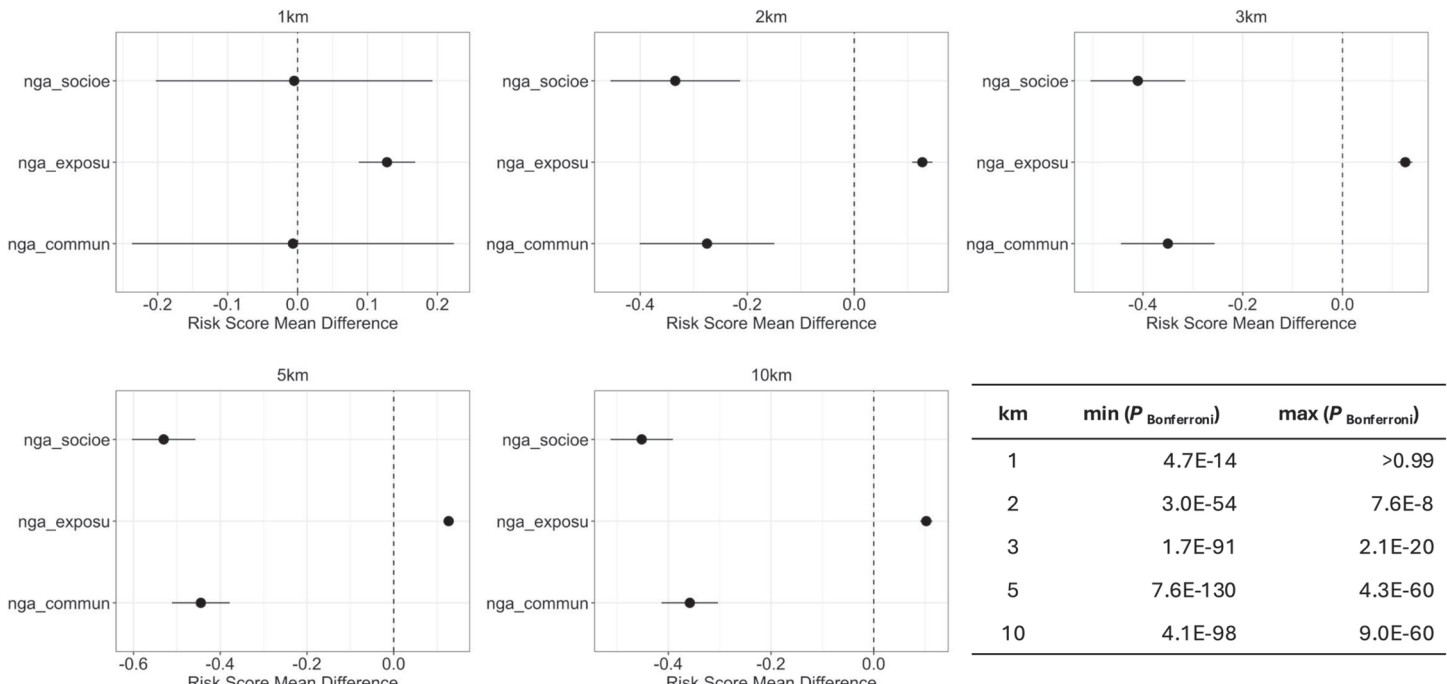

**Fig 13. Hypothesis 3:** The mean difference between "Class 1" (a school attack happens within $k$ kms. of a given school) and "Class 0" (no school attack happens within $k$ kms. of a given school) for $k \in \{1, 2, 3, 5, 10\}$ increases as the Socioeconomic Risk Score of the region of each school decreases (i.e. the school is in a wealthier neighborhood). Numerical values associated with each point estimate and 99% confidence interval can be found in Table S6, S5 Appendix E.

The testing of Hypothesis 3 yields more nuanced results than the previous hypotheses. We find that the mean differences between Class 1 and Class 0 with respect to both the Socioeconomic Risk Score and Communication Risk Score are negative. In other words, these risk scores are low for Class 1 and high for Class 0. This finding supports the notion that schools with a higher probability of attack tend to be located in wealthier areas (i.e. areas with low values of these risk scores). In contrast, we also find that the difference between the means of Class 1 and Class 0 in Exposure Risk scores is positive, suggesting that regions with poor sanitation and water are at greater risk of school attacks. One possibility might be that regions with poor sanitation and water tend to be in rural areas.

Based on our analysis of socioeconomic and communication risks, Boko Haram targets both wealthy areas for kidnapping, perhaps to garner good ransom payments, as well as poor areas, perhaps to obtain child soldiers and domestic workers and/or sex slaves. There is evidence supporting the hypothesis that Boko Haram generates revenue through kidnapping and ransom [64]. See also the assertion in [65], attributed to an unspecified 2021 National Geographic article, that ransoms were paid for 103 of the 276 girls kidnapped in Chibok, though official reports from the Nigerian Government at the time deny making ransom payments to Boko Haram in exchange for the release of the Chibok girls [66].

Finally, we hypothesize that Boko Haram targets schools in rural areas more frequently than urban areas when urban/rural areas are defined by population. There is a wide body of literature that has looked at the question of whether terrorist groups prefer urban or rural targets. For instance, [67] argues that "urban locations make attacks against civilian targets more likely, whereas rural areas increase the likelihood of attacks against the police and governmental targets". The question of urban vs. rural locations in the context of schools has not

been studied—and this is why we chose to study it below. [67] also argues that urban locations are attractive for the density of targets and for the ability of terrorists to move with relative anonymity—whereas in a rural region, outsiders may stand out. As population density seems linked to this hypothesis, we investigated a secondary hypothesis. This secondary hypothesis is based on the idea that urban areas can be further classified into urban clusters (areas of $> 50,000$ *people*) *and urban centers* (*areas of* $> 2,500$ *and* $< 50,000$ people) [68]. We expected that the less densely populated region would be targeted more frequently. Our hypotheses can be more formally stated as two questions:

**Hypothesis 4.** *(a) Does the frequency of school attacks differ between Rural vs. Urban (defined based on census-based population)?*
*(b) Considering non-rural schools only, does the frequency of school attacks differ between Urban Center and Urban Clusters?*

To formally address Hypothesis 4(a), we use the data of all schools to set up a $2 \times 2$ contingency table with respect to the categories of interest and compute the odds ratio (OR) and 99% CI. We also calculated statistical significance using the Chi-squared test with Yates' Continuity Correction [69]. The results are shown in Table 2. Appendix G in the Supplementary Material shows additional statistics for the $2 \times 2$ contingency table that go beyond odds ratios. These include PPV, NPV, Sensitivity, and Specificity.

Next, we restrict our data to a subset containing non-rural schools (Table 2B). We compare the proportion of school attacks in the more densely populated "urban centers" to the less densely populated "urban clusters" using the same statistical approach.

*Results and Analysis of Hypothesis 4.* Table 3 shows the results of our analysis. We find that the likelihood of school attacks in urban regions is 4.52 times higher than in rural regions. Limiting to only the schools in urban regions, the frequency of school attacks in urban centers is 9.42 times higher than in urban clusters. Table S7 in S7 Appendix G provides additional metrics for a more comprehensive evaluation and deeper insights into the analysis.

**Finding 4.** *These results suggest that Boko Haram targets schools in more densely populated areas (schools in urban centers are more likely to be targeted than schools in urban clusters, which, in turn, are more likely to be targeted than schools in rural areas).*

**Table 2. Contingency table setups for the hypotheses.**

|  | (A) | | (B) | |
|---|---|---|---|---|
|  | All schools, $n = 103,064$ | | Urban schools only, $n = 74,281$ | |
|  | Urban | Rural | Urban Center | Urban Cluster |
| **Attack=1** | 2386 (3.21%) | 210 (0.73%) | 2145 (5.79%) | 241 (0.65%) |
| **Attack=0** | 71,895 (96.79%) | 28,573 (99.27%) | 34929 (94.21%) | 36,966 (99.35%) |

**Table 3. Association analysis of school attacks and socioeconomic wealth.**

|  | Odds Ratio | 99% CI, lower | 99% CI, upper | P-value |
|---|---|---|---|---|
| **Urban** |  |  |  |  |
| (compared to Rural; all schools included) | 4.52 | 3.92 | 5.23 | $< 5 \times 10^{-115}$ |
| **Urban centers** |  |  |  |  |
| (compared to Urban Clusters; Rural schools excluded) | 9.42 | 8.23 | 10.81 | $< 1 \times 10^{-6}$ |

## Machine learning based analysis

In this section, we describe the results of machine learning-based analysis of whether Boko Haram would target a school. We note that our goal in this paper is not to develop new machine learning techniques, but to adapt them to gain insights about the risk that schools in Nigeria face. We report the results of two experiments:

1. How well can we predict the schools that Boko Haram will attack as we vary the granularity of prediction with $k \in \{1, 2, 3, 5, 10\}$ km?
2. Which features are the most impactful in making these predictions? The answer to this latter question sheds light on the factors that are most important in determining where Boko Haram targets its school attacks.

*Predictive Performance* Our data consisted of triples $(s, \vec{f_s}, \vec{dv_s})$ where $s$ is a school, $\vec{f_s}$ is the feature vector of length 15 associated with $s$, and $\vec{dv_s}$ is a vector of 5 dependent variables. The feature vector $\vec{f_s}$ consists of the 15 features shown in S1 Appendix A Table S1. The vector $\vec{dv_s}$ of dependent variables includes one dependent variable each for the $k = 1, 2, 3, 5, 10$ km variables. The value of the dependent variable is set to 1 for a school $s$ if Boko Haram attacked a school within $k$ km. of $s$. Otherwise, it is set to 0.

Machine learning classifiers fall into two broad categories: those based on deep learning and more traditional ones. We selected six of the most well-known traditional machine learning classifiers that have performed well in many other settings. The six traditional classifiers were Random Forest, Decision Trees, AdaBoost, Logistic Regression, Linear SVM, and Gaussian Naive Bayes. For instance, Random Forest classifiers have outperformed many other machine learning classifiers in many settings. On the deep learning side, we also tried Multi-Layer Perceptrons and Deep Neural Networks—the first because it is a foundational one and the second because it is extremely popular in the literature today. As a third deep learning model, we also tried a time-based Long Short-Term Memory (LSTM) neural network based model—but the results in this case (see S9 Appendix I) were truly abysmal and hence are not reported in the main body of the paper.

The features used by these classifiers are shown in S1 Appendix A. These features were selected on the basis of theories in social science from the fields of criminology, conflict studies, and sociology. The rationale for features related to distance to the nearest security installation is based on the social science theory that the presence of police stations deters crime [15,16]. The rationale for features related to the rural vs. urban nature of crime is based on the idea that urban crime may be more prevalent in some cases [22] and rural crime may be more prevalent in some cases [67]. The rationale for features based on socioeconomic characteristics of a region are based on the idea that poor neighborhoods (e.g. in the USA) are more likely to experience crime) than rich ones [20,21].

All our machine learning performance results were obtained via a standard 10-fold cross-validation protocol in which training/validation was done on 9 folds and testing was done on the 10th (holdout) fold. This was repeated 10 times for each classifier by varying the holdout fold in each of the 10 iterations and training/validating on the remaining 9 folds. Table 4 shows the Precision, Recall, F1 Score, and Area under the Receiver-Operation Characteristic Curve (AUC) for the best-performing classifier. AUC and F1 scores are single performance metrics—the latter combines Precision and Recall. For all $k$'s, AdaBoost generated the best results in both F1 Score and AUC. S2 Appendix B shows the detailed breakdown of the predictive performance of all the classifiers tested for all $k$'s.

The predictive performance results achieved by AdaBoost, our best classifier, are shown in Table 4. The results show that predictive performance is highly accurate when we consider

**Table 4. Summary of performance of the best machine learning classifier (AdaBoost).**

| $DV_k$ | Precision | Recall | F1 Score | AUC |
|---|---|---|---|---|
| 1 km. | 0.83 | 0.62 | 0.71 | 0.81 |
| 2 km. | 0.92 | 0.91 | 0.91 | 0.95 |
| 3 km. | 0.94 | 0.92 | 0.93 | 0.96 |
| 5 km. | 0.96 | 0.97 | 0.97 | 0.98 |
| 10 km. | 0.99 | 0.98 | 0.98 | 0.99 |

$k \geq 2$. In such cases, both precision and recall are above 90%. This means that when our AdaBoost models for $k \geq 2$ suggest that a school will be attacked, there is a high probability that a school within a few kilometers of it (i.e. $k$ kms.) will in fact be attacked, over 91%. Moreover, the recall is also high for $k \geq 2$, meaning that in the case of schools that were in fact attacked, our algorithms are able to correctly predict this.

Throughout this analysis, we see that as $k$ increases from 1 to 10, all performance metrics also improve. When we consider $k = 1$, both precision and recall drop. This suggests that predicting at the 1 km. spatial granularity is much more challenging than at the granularity of 2 km. or more. Yet, precision is still 83% suggesting that when AdaBoost predicts a school will be attacked, there is an 83% probability that prediction is correct. But recall drops dramatically to 62% for $k = 1$, compared to over 91% for $k = 2$ and higher.

The performance metrics are also illustrated in Fig 14, where the AdaBoost Receiver Operating Characteristic (ROC) curves for all $k$'s are plotted. These curves demonstrate the trade-off between the true positive rate and the false positive rate across various thresholds. S8 Appendix H contains the confusion matrices of the AdaBoost model across all $k$'s. As $k$ decreases, the model's ability to predict positive instances also decreases, suggesting that it performs better on a larger spatial scale. To ensure reproducibility, it is important to note that the model was run using version 1.1.2 of the Scikit-Learn library.

*Finding the Most Impactful Features* To identify the features that have the biggest impact on predictive accuracy, we used standard ablation testing. In ablation testing, we drop one feature at a time from the overall set of 15 features considered (cf. Table S1 in S1 Appendix A), and measure the reduction in predictive performance (F1 score). The most important feature, when dropped, leads to the greatest reduction in predictive performance. Suppose we now drop the most important feature (after identifying it). We need to recompute the drop in predictive performance for each remaining feature to find the second most significant feature. This process is repeated to find the third most important feature, and so forth. Table 5 shows the three most important features (by rank) that our best-performing model (AdaBoost) used, as we vary $k$. The rows of this table show the features and the columns show $k$. For instance, the row "dist. to 3rd closest security installation" suggests that this is the most important feature for the dependent variable with $k = 5$ km. and the second most important feature for the $k = 2$ km. dependent variable.

**Finding 5.** *The distances to nearby security installations form the most important type of variable when predicting which schools will be attacked. The exact rank of which feature is more important in predicting importance of a feature for the different $k \in \{1, 2, 3, 4, 5\}$ km. variations of the dependent variable are perhaps not important. What is important is that the the features about proximity of the school to the nearest, 2nd, 3rd, 4th, and 5th nearest security installations consistently end up being highly ranked (in the top 3) across many of the variations of $k$. That said, it would be interesting for future work to understand why the ranks turn out the way they do for different variations of $k$. This reinforces the findings in Hypothesis 2 which we had*

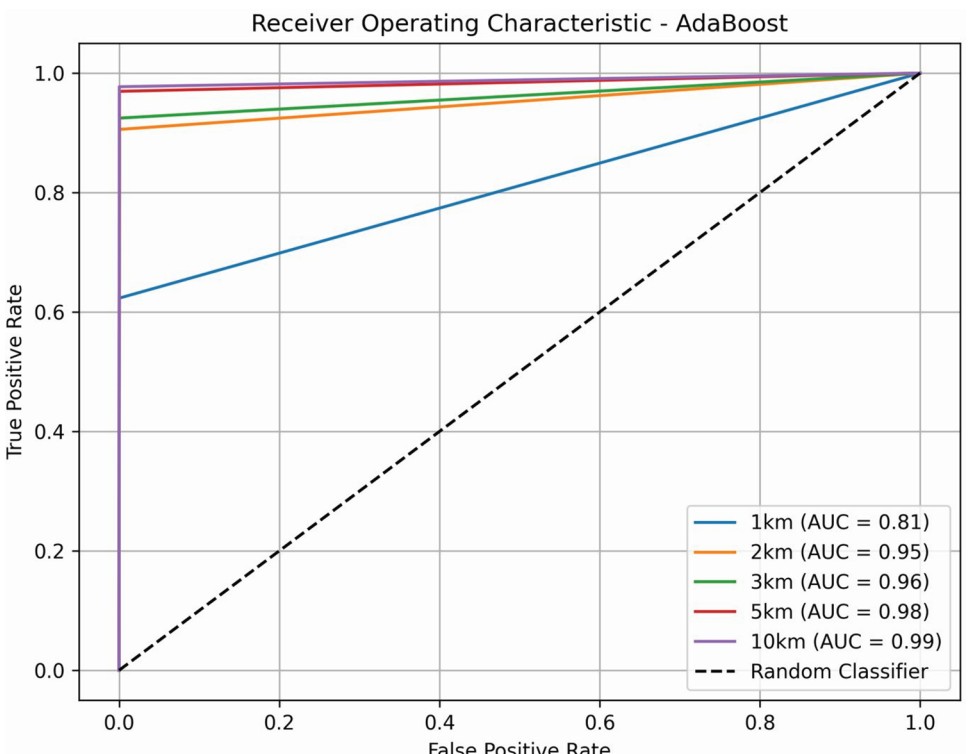

**Fig 14.  Receiver operating characteristic curve for AdaBoost.**

**Table 5. Ranks of the importance of features (rows) as we vary the dependent variable $k$ (columns). Determined using ablation testing with our best-performing classifier, AdaBoost.**

| Feature name | $k = 10$ | $k = 5$ | $k = 3$ | $k = 2$ | $k = 1$ |
|---|---|---|---|---|---|
| Attacks within 10 kms. of school | 1 | | | | |
| Dist. to closest security installation | 2 | | 3 | 1 | 3 |
| Dist. to 2nd closest security installation | | | 2 | 3 | 1 |
| Dist. to 3rd closest security installation | | 1 | | 2 | |
| Dist. to 4th closest security installation | | 3 | | | |
| Dist. to 5th closest security installation | 3 | 2 | 1 | | 2 |

*explored earlier, showing that the presence of security forces near schools is a powerful deterrent to Boko Haram attacks on schools.*

*Multivariate Decision Trees.* In addition to high-quality predictions generated by AdaBoost, our Decision Tree classifiers also showed high performance with F1 scores of 0.38, 0.79, 0.85, 0.78, 0.81 for the $k = 1, 2, 3, 5, 10$ km dependent variables. Except for the $k = 1$ case (where, as we have seen earlier, our predictive ability is weak), these F1 scores are all over 0.75 and hence considered quite strong. A notable advantage of decision trees is that they are easily explainable. Fig 15 shows part of our decision tree for the $k = 1$ dependent variable. Showing the entire decision tree is challenging as it causes the fonts to become very small and unreadable. Hence, we chose to focus on the parts of the decision tree that are relevant to our claims. The bold blue line shows the following hypothesis. Let $S$ be the set of schools that satisfy the following logical conditions: the number of total attacks

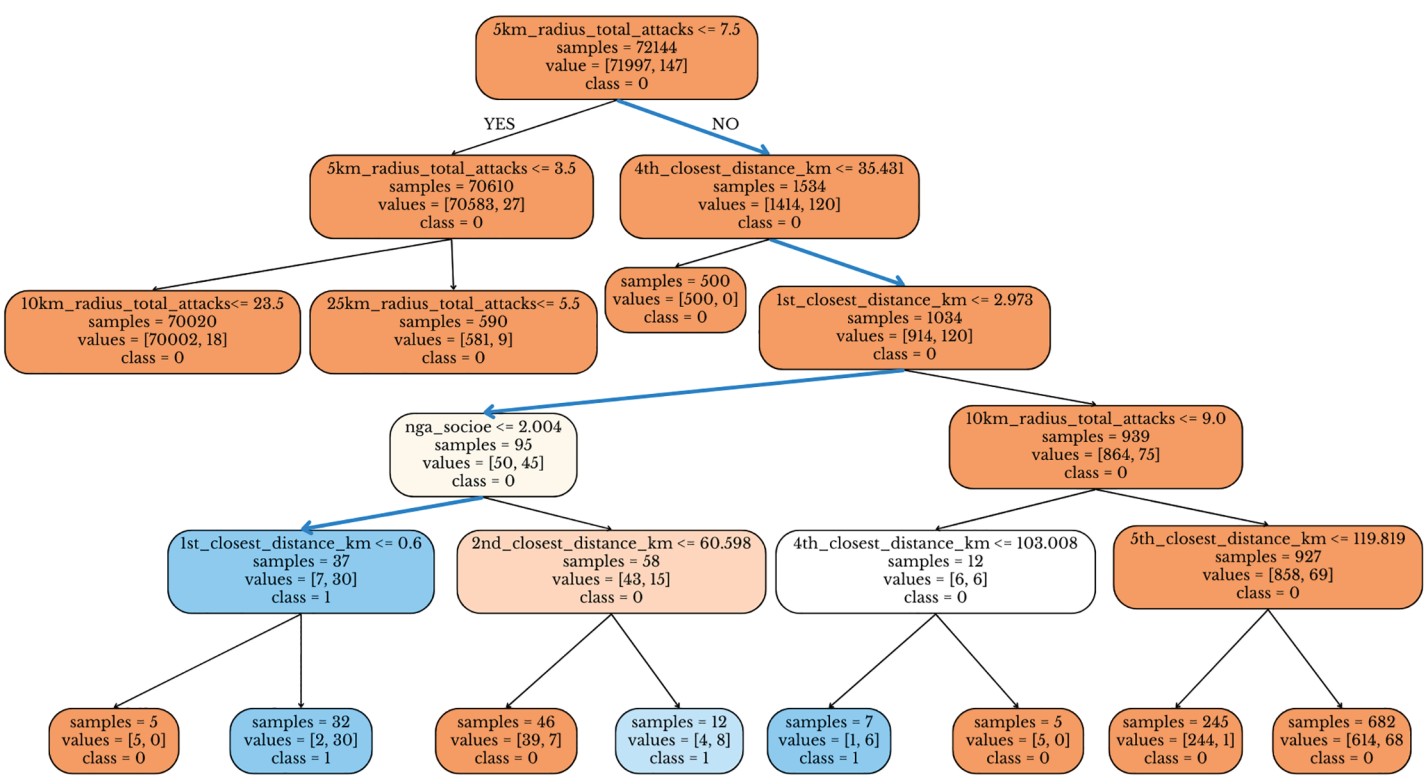

**Fig 15. Relevant Part of the Decision Tree Extracted for k=1 Dependent Variable.**

within 5km of the school exceeds 7.5 and the distance to the 4th closest security installation to the school exceeds 35.431 km. and the distance to the closest security installation to the school is less then 2.973 km. and the socioeconomic score of the ward to which the school belongs is less than 2.004. There were 37 schools satisfying this condition, 30 of which experienced Boko Haram attacks. This leads us to formulate the hypothesis that schools in $S$ (schools satisfying this logical condition) are far more likely to experience a kidnapping attack by Boko Haram than schools in $\bar{S}$. Similarly, the hypotheses can be generated from all of the decision trees that we have learned. S3 Appendix C presents all the decision trees we have extracted for $k = 2, 3, 5, 10$ km. cases.

## Multivariate machine learning inspired statistical inferences of Boko Haram school attacks

Though AdaBoost is the best performing predictive model, Decision Trees provide explanations that are easy to interpret. We therefore looked at the decision trees (for the $k = 1, 2$) cases. Fig 15 shows the decision tree we derived for the $k = 1$ case. Even though predicting which schools will be attacked is least effective when $k = 1$ as shown in Table 4, this decision tree still allows us to come up with some interesting hypotheses. As we will show shortly below, we are able to get strong experimental results.

In this section, we discuss new hypotheses based on the decision tree models described in the previous section. We then formally address these hypotheses using a maximum likelihood-based statistical approach.

The paths we select in our decision trees are highlighted in the hypotheses below. For example, the path highlighted in blue in Fig 15 reflects a condition which, when satisfied by a school, leads to a 30/32 = 93.75% probability that the school will experience an attack. In contrast, the chance of a school being attacked (unconditionally) is $76/103064 = 7.37x10^{-4}$ which is far less than 1%. Thus, even the decision trees for the $k = 1$ case can shed important light on conditions that, when satisfied by a school, suggest a high likelihood of an attack on that school. S3 Appendix C shows the decision trees for the $k = 2, ... , 5$ cases which also have similarly interesting paths.

We first investigate the hypotheses inspired by the example in Fig 15.

**Hypothesis 5.** *Let S be the set of schools that satisfy the following logical and condition. Each school s ∈ S must be such that there are:*

1. *at least 7.5 attacks within a 5 km radius of the school and*
2. *the distance of the 4th closest security installation to s is over 35.431 km and*
3. *the distance to the nearest security installation to s is 2.973 km or less and*
4. *the socioeconomic risk score for the school is less than 2.004.*

*Let S̄ be the set of all other schools (i.e. those that do not satisfy the above condition). We hypothesize that schools in S are far more likely to experience a Boko Haram kidnapping attack within 1 km distance by Boko Haram than schools in S̄.*

There were 37 schools in *S* and 30 of them experienced attacks. Table 6 shows the contingency table for testing the hypothesis. We use an odds ratio (OR) as a representation of how many times, on average, school attacks would occur in *S* compared to *S̄*.

- *OR* > 1 There was an enrichment of school attacks in set *S*
- *OR* < 1 There was a depletion of school attacks in set *S*. (or equivalently, there was an enrichment of school attacks in set *S̄*.
- *OR* = 1 (i.e. null value) indicates no difference between set *S* and *S̄*.

As before, we used the P-values from the Chi-squared test with Yates' continuity correction [69] to determine statistical significance.

We similarly set up a contingency table for each of the remaining six decision-tree hypotheses. Table 7 shows that in general:

- all OR point estimates are large and positive
- none of the 99% confidence intervals (CI) contain the null value of 1
- all Chi-squared test P-values are well below $10^{-3}$

These results serve as strong evidence to support Hypothesis 5. Furthermore, additional metrics that enhance the understanding of DT-based statistical inference studies such as PPV, NPV, Sensitivity, and Specificity, are detailed in Table S8, found in S7 Appendix G.

**Table 6. Example of contingency table setup for testing Hypothesis 5.**

|              | Set *S*        | Set *S̄*         |
|--------------|----------------|------------------|
| **Attack = 1** | 30 (81.08%)  | 117 (0.16%)      |
| **Attack = 0** | 7 (18.92%)   | 71990 (99.84%)   |

**Table 7. Results of statistical evaluation of the decision tree hypotheses.**

| km | Hyp. No. | Attacks Set $S$ | | Attacks Set $\bar{S}$ | | Odds Ratio | 99% CI, lower | 99% CI, upper | Chi-squared P-value |
|---|---|---|---|---|---|---|---|---|---|
| | | Count | % | Count | % | | | | |
| 1 | 5 | 30 | 81.08 | 117 | 0.16 | 2632.5 | 1122.2 | 8192.0 | 0.0002 |
| 1 | 6 | 30 | 93.75 | 117 | 0.16 | 7766.5 | 2409.4 | 4.5e15 | 0.0002 |
| 2 | 7 | 50 | 89.29 | 358 | 0.50 | 1616.9 | 685.9 | 4300.3 | 0.0002 |
| 2 | 8 | 50 | 94.34 | 358 | 0.50 | 3197.0 | 1050.3 | 16384.0 | 0.0002 |
| 2 | 9 | 122 | 82.99 | 286 | 0.40 | 1234.1 | 753.6 | 2015.4 | 0.0002 |
| 2 | 10 | 118 | 93.65 | 290 | 0.40 | 3320.5 | 1789.3 | 8192.0 | 0.0002 |
| 2 | 11 | 118 | 95.93 | 290 | 0.40 | 5658.4 | 2409.4 | 16384.0 | 0.0002 |

**Hypothesis 6.** *Let S be the set of schools that satisfy the following logical condition. Each school s ∈ S must be such that there are:*

1. *between 7.5 and 80.5 attacks occurred within a 5 km radius of the school and*
2. *the distance of the 4th closest security installation to s is over 35.431 km and*
3. *the distance to the nearest security installation to s is between 0.6 and 2.973 km or less and*
4. *the socioeconomic risk score for the school is less than 2.004.*

*Our hypothesis is that schools in S are far more likely to experience a kidnapping attack by Boko Haram than schools in S̄.*

Notably, Hypothesis 6 differs slightly from Hypothesis 5 in two respects. Conditions (1) and (2) each now include a lower bound which is not present in Hypothesis 5. Formally testing this hypothesis, we determine that the frequency of attack in set *S* is over 7,700 times ($99\% CI = 2409.4 - 4.5 \times 10^{15}$) higher than set $\bar{S}$ with a highly significant $chi^2$ P-value of 0.0002.

Table 8 contains some additional hypotheses suggested by the decision tree for *k* = 2. We do not discuss these in detail here for space reasons. S11 Appendix K provides a comprehensive description and interpretation of these hypotheses for k=2, as detailed in Table 7.

Taken together, our statistical evidence supports the conclusion that schools satisfying the conditions to be in *S* are far more likely to experience attacks by Boko Haram than schools that do not.

## Limitations

Our work has several limitations.

First, like many open source projects, our effort is limited by the data that we were able to collect. The ACLED dataset [6], though excellent and widely used, may have missed some school attacks and other types of attacks by Boko Haram. Likewise, the GRID3 Data Hub includes data about school locations, security installations, and socioeconomic data about most of Africa - but may likewise be incomplete. We have tried to augment GRID3 with data from another widely used system, OpenStreetMaps, to the extent possible.

Second, it is possible that some attacks on schools have not been reported, e.g. if local organizations were able to quickly raise the alarm and stop the attack and/or if local leaders were able to quickly negotiate a resolution of the situation with Boko Haram. Such attacks may not be included in the ACLED data.

Third, we predict the risk of attack on individual schools. Our prior book [27] predicts *when* an attack will occur on a given *type* of target. Types of targets may include security

**Table 8. In this table, each hypothesis consists of 3–5 preconditions whose conjunction (logical and) implies a school attack.**

| Hyp. No | Preconditions | Parameter | Requirement |
|---|---|---|---|
| Hyp. 7 | 1 | Number of attacks within 5 km radius | Between 7.5 and 80.5 |
| | 2 | Number of attacks within 25 km radius | ≤ 17.5 |
| | 3 | Distance to nearest security installation | ≤ 2.73 km |
| Hyp. 8 | 1 | Number of attacks within 5 km radius | Between 7.5 and 80.5 |
| | 2 | Number of attacks within 25 km radius | ≤ 17.5 |
| | 3 | Distance to nearest security installation | ≤ 2.73 km |
| | 4 | Communication risk score for the school | ≤ 1.022 |
| Hyp. 9 | 1 | Number of ttacks within 5 km radius | > 80.5 |
| | 2 | Socioeconomic risk for the school | ≤ 2.283 |
| | 3 | Number of attacks within 25 km radius | > 88.5 |
| Hyp. 10 | 1 | Number of attacks within 5 km radius | > 80.5 |
| | 2 | Socioeconomic risk for the school | ≤ 2.283 |
| | 3 | Number of attacks within 25 km radius | > 88.5 |
| | 4 | Distance to 4th nearest security installation | > 116.098 km |
| Hyp. 11 | 1 | Number of attacks within 5 km radius | > 80.5 |
| | 2 | Socioeconomic risk for the school | ≤ 2.283 |
| | 3 | Number of attacks within 25 km radius | > 88.5 |
| | 4 | Distance to the 4th nearest security installation | > 116.098 km |
| | 5 | Number of attacks within 10 km radius | > 84.5 |

**Interpretation:** Let $S$ be the set of schools that satisfy the above conditions. Each school $s \in S$ meets these criteria. Our hypothesis is that schools in $S$ are far more likely to experience a kidnapping attack by Boko Haram than schools in $\bar{S}$.

installations, transportation targets, and of course, schools. This paper focuses on predicting risk of attack on *specific* targets, i.e. specific schools, not schools in general. Ideally, we would like to *simultaneously* predict both the time and the location of a school attack. However, simultaneous prediction at such fine-grained granularity is challenging. An important future work would extend the results here to predict the approximate time frames of attacks on specific schools. For now, the results show risks to schools in this paper and the "when" attacks might happen could be predicted using the methods in [27] as a starting point.

Fourth, Boko Haram is constantly evolving its tactics. This requires that the data be updated periodically and the models relearned. Fortunately, updating the data (e.g. on a monthly or quarterly basis) is not challenging using the same data sources used in this paper that are typically updated with a few weeks of delay. This allows for a constant update of the risk maps.

Fifth, machine learning algorithms can overfit the data. One reason we did both a rigorous statistical and machine learning analysis is to see if both sets of techniques provide similar results. Our most significant finding, namely that the distance of the nearest security installation from a school is linked to the risk of an attack on the school by Boko Haram is validated by both sets of techniques. Hence, we feel confident about this finding.

Sixth, there is the question of explainability. We are able to explain many findings quite well. Some of the decision tree findings are more complex to explain. For instance, if we look at Hypothesis 5, the condition used to define the sets $S$ and $\bar{S}$ depend on the distance of the school being studied from the nearest security installation and the 4th nearest security installation. This may look a little odd. We can think of this as expressing a range saying that the nearest security installation should be really close (less than 0.6 km) in the case of Hypothesis 5) and there are at least two other other security installations (i.e. the 2nd and 3rd

closest ones) within a distance of 35.431 kms. This might suggest that the exact locations of the 2nd and third closest security installations doesn't matter as long as they are within the desired distance.

Finally, there is the question of ethics. All the data used in this study has been ethically sourced, and no human data or personally identifying information (PII), whatsoever, is used in this study. The subject of study, school attacks, is at the school level, not at the level of individuals. Our results show that the absence of security installations close to schools is linked to increased probability of attacks by Boko Haram on schools - which suggests an increased security presence not too far from schools. This finding has been validated both statistically and through the ablation tests applied to our machine learning models. As such, it is robust. Please note that we are not suggesting any security presence inside schools, but within a few kilometers of schools which doesn't seem overly intrusive.

## Conclusion and recommendation

Since the start of the Boko Haram insurgency about 15 years ago, the group has carried out numerous deadly attacks including assassinations, attacks on markets, attacks on government buildings and security installations, and even attacking UN headquarters in Abuja. Despite numerous efforts to deter these attacks, the group remains active as of March 2024.

Protecting children is a strong imperative for society. The impact of school attacks on the boys and girls who are kidnapped is horrific. Not only do those who are abducted suffer the horrors of domestic servitude, sexual slavery, torture, enrolment as suicide bombers and child soldiers, and other forms of abuse, even those who escape these abductions are left with a deep reluctance to return to schools for fear of kidnapping. In the absence of a quality education, they may be doomed to a life of poverty.

In this work, we present a data-driven investigation of Boko Haram's attacks on schools with rigorous statistical profiling and machine-learning analytics. Notably, we assemble a novel dataset spanning almost 14 years (available as part of our Supplementary Material). Using this data, our work leverages both statistical inference and machine learning to characterize Boko Haram's school attacks in several novel aspects. First, we develop a set of univariate hypotheses based on a triad of factors: for a school to be vulnerable, (i) there must be Boko Haram *Activity* in the vicinity of the school (ii) *Security Presence* in the area around the school must be weak, and (iii) the *Socioeconomic* conditions prevalent in the vicinity of the school must be poor. While our statistical analyses validate the first two of these hypotheses, our findings on the third hypothesis are less straightforward. Wealthy areas are targeted (perhaps to gain ransom payments)—but so are poor areas (perhaps to capture sex slaves and child soldiers). Second, we apply advanced machine learning algorithms to predict which schools are at risk of attack by Boko Haram. Our algorithms perform well: when we consider a prediction to be correct when a school within 2 km. of the considered school is attacked, then we obtain both precision and recall exceeding 90%.

Third, through *Ablation Tests*, we find that the distance to nearby security installations is the single, most important risk factor of experiencing a school attack. In particular, the finding implies that increasing security presence around vulnerable schools will likely have a deterrent effect. Fourth, our machine learning models motivate the design of seven hypotheses that can be statistically addressed. We formally articulate and validate these hypotheses linking a diverse set of independent variables relating to the activity of Boko Haram, security presence, and socioeconomic conditions of a given region in Nigeria.

Finally, a major contribution of our work is the geospatial mapping of school vulnerability (Figs 1 and 2). Fig 1 displays the risk of school attacks for each and every school in Nigeria, while Fig 2 focuses on parts of Northeast Nigeria and shows the risk of school attacks there.

*Recommendation.* A major recommendation of this paper to both international donors and the Nigerian government is the need for a commitment to reducing the distance between vulnerable schools and nearby security installations. Our findings show clearly that the presence of security installations near schools is the single biggest factor in whether schools are attacked by Boko Haram. This needs to be an urgent priority for the Nigerian Government.

International donors and Nigeria's security partners should also take note. As an example, a January 2024 U.S. State Department report [70] mentions millions of dollars of aid to Nigeria and hundreds of millions of dollars of weapons sales. As stated in [71] in a *Foreign Policy* article, such weapons sales may not solve Nigeria's security problems. We recommend that at least some amount of such financial aid efforts focus, over time, on creating police stations near high-risk schools such as those shown in Figs 1 and 2.

Because of widespread allegations of abuse by Nigeria security forces, there is also a need for improved training and oversight of such forces.

The results generated using our best performing algorithm, AdaBoost, show that the single biggest factors involved in school attacks is the absence of security installations near many vulnerable schools. The results therefore suggest that in order to reduce the number of school attacks by Boko Haram, there needs to be a greater security presence in Nigeria, both in the Northeast of the country and in some Northern states like Kano State. Foreign aid and domestic investments intended to strengthen security in these regions must be accompanied with detailed analyses on exactly where such security installations must be located in order to maximize the protection that these schools and schoolchildren deserve. Specifically, the results in this paper provide a risk score for each school which can be used, in conjunction perhaps with facility location algorithms [72], to identify the best locations for such security installations. The creation of these new security installations must be accompanied with highly visible patrols whose routes vary daily so that a deterrent effect can be achieved. Till such interventions occur with the force needed, a stable situation will be hard to reach.

## Supporting information

**S1 Appendix A. Independent features calculated for every school record and their description.**
(PDF)

**S2 Appendix B. Performance Metrics of Various Classifiers Across Different Dataset Variations.**
(PDF)

**S3 Appendix C. Decision Tree Graphs.**
(PDF)

**S4 Appendix D. Ward's Wealth and Residential Area Type.**
(PDF)

**S5 Appendix E. Supplementary Tables for the Forest Plots.**
(PDF)

**S6 Appendix F. OverPass API Queries for Military Installations.**
(PDF)

**S7 Appendix G. Additional Metrics for ML-Inspired Statistical Analyses.**
(PDF)

**S8 Appendix H. AdaBoost Confusion Matrices.**
(PDF)

**S9 Appendix I. Time-Series Analysis.**
(PDF)

**S10 Appendix J. Description of Detailed Statistical Methods.**
(PDF)

**S11 Appendix K. Additional Decision Tree - Inspired Statistical Analysis.**
(PDF)

**S1 Data. Hypotheses Test Data.**
(ZIP)

**S2 Data. ML Data.**
(ZIP)

## Acknowledgements.

We thank Dan Byman, Aaron Mannes, Chiara Pulice, and Marco Postiglione for detailed comments on earlier versions of this manuscript.

## Author contributions

**Conceptualization:** V.S. Subrahmanian.

**Data curation:** Lirika Sola.

**Formal analysis:** Lirika Sola, Youdinghuan Chen, V.S. Subrahmanian.

**Methodology:** V.S. Subrahmanian.

**Project administration:** V.S. Subrahmanian.

**Software:** Lirika Sola.

**Supervision:** V.S. Subrahmanian.

**Validation:** Lirika Sola, Youdinghuan Chen.

**Visualization:** Lirika Sola, Youdinghuan Chen.

**Writing – original draft:** Youdinghuan Chen, V.S. Subrahmanian.

**Writing – review & editing:** Youdinghuan Chen, V.S. Subrahmanian.

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
