## [Decision Letter · Decision Letter 0]

20 Dec 2024

PONE-D-24-14163A Quantitative Geospatial Analysis of the Risk that Boko Haram Will Target a SchoolPLOS ONE

Dear Dr. Subrahmanian,

Thank you for submitting your manuscript to PLOS ONE. After careful consideration, we feel that it has merit but does not fully meet PLOS ONE’s publication criteria as it currently stands. Therefore, we invite you to submit a revised version of the manuscript that addresses the points raised during the review process.

 Please accept our apologies for the delay in issuing an editorial decision. Unfortunately, we have to search for a new Academic Editor several times since you submitted your manuscript. 

The manuscript has now been evaluated by six reviewers, and their comments are available below.This is a large number of reviewers. However, we have had difficulty securing reviewers with the relevant expertise to assess the topic and methods of your study. Although all the reviewers have brought their own perspectives to bear, and all offer constructive criticism, please pay particular attention to the comments made by reviewers 2 and 5. Could you please revise the manuscript to carefully address the concerns raised?

We look forward to receiving your revised manuscript.

Kind regards,

Steve Zimmerman, PhD

Senior Editor, PLOS One

Journal Requirements:

3. We note that Figures 1,2 and 5 in your submission contain [map/satellite] images which may be copyrighted. All PLOS content is published under the Creative Commons Attribution License (CC BY 4.0), which means that the manuscript, images, and Supporting Information files will be freely available online, and any third party is permitted to access, download, copy, distribute, and use these materials in any way, even commercially, with proper attribution. For these reasons, we cannot publish previously copyrighted maps or satellite images created using proprietary data, such as Google software (Google Maps, Street View, and Earth). For more information, see our copyright guidelines: http://journals.plos.org/plosone/s/licenses-and-copyright.

1. You may seek permission from the original copyright holder of Figures 1,2 and 5 to publish the content specifically under the CC BY 4.0 license.

Reviewers' comments:

Reviewer's Responses to Questions

**Comments to the Author**

1. Is the manuscript technically sound, and do the data support the conclusions?

Reviewer #1: Yes

Reviewer #2: Partly

Reviewer #3: Yes

Reviewer #4: Yes

Reviewer #5: Partly

Reviewer #6: Yes

2. Has the statistical analysis been performed appropriately and rigorously? 

Reviewer #1: Yes

Reviewer #2: No

Reviewer #3: Yes

Reviewer #4: Yes

Reviewer #5: Yes

Reviewer #6: Yes

3. Have the authors made all data underlying the findings in their manuscript fully available?

Reviewer #1: Yes

Reviewer #2: Yes

Reviewer #3: Yes

Reviewer #4: Yes

Reviewer #5: Yes

Reviewer #6: Yes

4. Is the manuscript presented in an intelligible fashion and written in standard English?

Reviewer #1: Yes

Reviewer #2: Yes

Reviewer #3: Yes

Reviewer #4: Yes

Reviewer #5: No

Reviewer #6: Yes

5. Review Comments to the Author

Reviewer #1: The article is interesting and could be used for public awareness. The article needs further improvement. My comment are;

1. Citation is missing in 'A similar claim was repeated by Nigeria’s Information Minister in October 2019()'.

2. Give a predictive AI based solution for the problem mentioning the period it might take for a stable situation

Reviewer #2: In the manuscript, utilizing a geospatially tagged data set, the coauthors identified factors affecting attacks on schools, implemented machine learning models to quantify the likelihood that a school will be at risk of a Boko Haram attack and finally concluded with a policy recommendation. The work presented is dedicated to an interesting and important topic. I have multiple concerns are as follows:

1. Order of Figure 3a vs. 3b is wrong. And in the caption “2023 numbers only run …” 2023 seems redundant?

2. In section Socio-Economic Data, there is no clear definition of “risk score”. How it derived from the data? Whether it’s valid or not?

3. Figure 6 and 8 are switched? Figure 6, 7, 8 are with low resolution.

4. In the sentence below Table 1, Table 1B summarized the non-urban schools or urban schools?

5. In the machine learning based analysis section, It seems like the models were only fitted in training set but not validation set nor testing set. It may lead to overfit problem. And prediction accuracy in training set is not valid for prediction in future events.

6. In line 446, there are two “that”.

7. In Figure 9, results for k=1 were shown. How connect the results with results in Table 3 where models for k=1 provided worst performance? Brief discussion is welcome.

8. For section “Multivariate Machine Learning Inspired Statistical Inferences of Boko Haram School Attacks”, using odds ratio assessing the hypothesis is not justifiable enough. Other assessment such as sensitivity, specificity, PPV, NPV, accuracy, error rate, …, could be provided in addition to OR.

9. Figure 1 & 2 displays the risk of school attacks. How those risk estimated? Which method was used?

Reviewer #3: The research article presents a quantitative geospatial analysis of the risk factors associated with Boko Haram's targeting of schools in Nigeria over nearly 14 years, utilizing statistical inference and machine learning techniques. It identifies three main vulnerability factors: proximity to Boko Haram activity, weak security presence, and socioeconomic conditions. The study finds that both wealthy and poor areas are targeted for different reasons—wealthy areas for ransom and poor areas for capturing sex slaves and child soldiers. Advanced machine learning models, particularly the AdaBoost classifier, achieved an F1 score of 0.85 in predicting attacks within 2km of schools, with key predictive features including distance to security installations and previous attacks. The findings underscore the critical role of security presence in deterring attacks and highlight the complex interplay between socioeconomic factors and risk. Overall, this research provides valuable insights for policymakers and security forces, suggesting that enhanced protection strategies could be developed based on the identified risk factors and predictive models. However, there are some concerns, listed below:

1. The authors should try to incorporate time series analysis or recurrent neural networks to better capture the temporal dynamics of conflict evolution over time.

2. The authors should remove the words first ever from abstract and conclusion.

3. Clarify the rationale behind choosing specific machine learning algorithms.

4. Provide more details on data preprocessing and feature selection methods.

5. Explain how the 2km threshold for predictions was determined.

6. All the tables should include axis labels.

7. It would be helpful with provide visualizations (e.g., ROC curves, confusion matrices) to illustrate model performance.

8. Address potential biases in the dataset and discuss their implications

9. Provide a separate methods section on statistical methods and analysis

Reviewer #4: More details about the data sources, particularly the socioeconomic data and the locations of security installations, would improve the manuscript. Reproducibility depends on transparency on data reliability and potential biases in data sources.

Describe any imputation methods that were employed and include a detail on how any missing data was handled. It is also necessary to address the handling of outliers, especially in socioeconomic and activity factors.

The manuscript would benefit from a brief description of how insights from the AdaBoost model and decision trees could be applied in a practical setting, even if the research highlights the significance of features in prediction.

By employing quantitative techniques to examine the risk factors associated with Boko Haram's attacks on schools, this paper tackles a pressing and important subject. It makes a significant contribution to the fields of geospatial analysis and counterterrorism studies.

Despite their strength, machine learning models have drawbacks. Talk about possible drawbacks such as the model's tendency to overfit historical data, particularly in light of Boko Haram's changing strategies, and how it might apply to attacks in the future or in other regions.

Since predictive models can impact real-world security decisions, address any ethical considerations related to this research, such as the potential consequences of model inaccuracies.

Reviewer #5: The paper A Quantitative Geospatial Analysis of the Risk that Boko Haram Will Target a School submitted to PLOSOne makes a number of interesting contributions that empirically, theoretically and practically relevant in the study of conflict and international development. The paper develops a one-of-a-kind dataset on Boko Haram (BH) attacks with substantial contextual information. It is a step up from widely-used ACLED. It then develops a set of spatial measures to examine hypotheses about mechanisms that may be involved in BH attacks. There is reasonable observational evidence that the level of general BH activity in the area and proximity to security (e.g., police or military facilities) are associated with the likelihood of attack on a school. If the patterns of association have a causal underpinning, then the authors point to useful policy recommendations.

Overall, the paper may be acceptable for publication in PLOSOne with substantial revisions. I offer the following suggestions in the hope that they help the authors revise the paper for resubmission.

The general points to consider are:

1. The paper could use a careful editing for language.

2. The story gets lost in a relatively large number of tests and separate analyses that are not fully described. The authors might consider slimming down the manuscript, concentrate on solid narrative development, and move “nice to have but not essential” testing to an integrated Supplementary file.

3. The main measure is a bit confusing. I have a hard time visualizing what it means to measure school attacks within 10 km against BH activity within 50 km. What if there is more than one school within that 10 km radius and one of them is attacked? I think a visualization of the method would go a long way to better supporting the findings.

The remainder of my comments are in order of presentation and mix both larger issues with some smaller ones.

1. Pg. 2, lines 16-17, is a very powerful statement. It would punch even harder if it ended with a something like "...all of this impact arises from a group with an estimated N number of members/fighters." Bring home the point that asymmetry can still produce huge harm.

2. Pg. 2, lines 43-53. This is small, but I would hang the activity hypothesis first on "routine activities theory" with a secondary pointer to "journey-to-crime". The basic idea is that areas where BH are operating have established some type of (operational) routine. Targets that fall near or within that routine are easier to accommodate than ones that are outside of the routine.

@article{cohen1979rat,

author = {Cohen, L. E. and Felson, M.},

title = {Social change and crime rate trends: A routine activity approach},

journal = {Am Sociol Rev},

volume = {44},

pages = {588-608},

DOI = {10.2307/2094589},

year = {1979},

type = {Journal Article}

}

3. Pg. 3, lines 61-62. If you want an explanation for "why" this might be, you could argue that the police stations serve as a deterrent where BH fighters perceive a greater likelihood of getting caught if operating near those spaces.

@article{RN5221,

author = {Nagin, Daniel S.},

title = {Deterrence in the Twenty-First Century},

journal = {Crime and Justice},

volume = {42},

pages = {199-263},

DOI = {10.1086/670398},

year = {2013},

type = {Journal Article}

}

@article{RN2912,

author = {Loughran, Thomas A. and Paternoster, Raymond and Piquero, Alex R. and Pogarsky, Greg},

title = {On Ambiguity in Perceptions of Risk: Implications for Criminal Decision Making and Deterrence},

journal = {Criminology},

volume = {49},

number = {4},

pages = {1029-1061},

DOI = {10.1111/j.1745-9125.2011.00251.x},

year = {2011},

type = {Journal Article}

}

4. Pg. 3, lines 65-66. This probably directly relates to so-called social disorganization theory in criminology. A key citation is (though there are thousands of others too):

@article{RN3950,

author = {Bursik Jr., Robert J.},

title = {Social Disorganization and Theories of Crime and Delinquency: Problems and Prospects},

journal = {Criminology},

volume = {26},

number = {4},

pages = {519-552},

DOI = {10.1111/j.1745-9125.1988.tb00854.x},

year = {1988},

type = {Journal Article}

}

5. Pg. 3, line 68. Hypothesis 4 is not particularly clear to me. I think the authors are looking forward to their results.

6. Pg. 6, lines 201-2002. There are a few years with no recorded attacks. Is this because there were no attacks or because of other potential issues with data acquisition?

7. Pg. 6, lines 210-211. When are schools on break? “While non-school attacks are more or less uniformly distributed over the 12 months of the year, school attacks occur less frequently in January, August, and December.”

8. Pp. 6-8. Overall, a tabular summary of data might be more efficient use of space.

9. Pg. 8, lines 276-277. The core measure used in the paper needs better explanation. It is not clear if this is a measure that the authors have developed themselves or if it has a source in the literature. There are a couple of ways to understand the statement “a school attacked happened within k ={1,2,3,5,10} km”. This could mean that there are, say, 10 students from school s. They weren't kidnapped directly from the school s but at a location k-kilometers away from the location of s. Or, there is a school s_i and a school s_j. They are 4.3km apart. If schooI s_i is attacked then school s_j also is considered attacked within the 5km distance band of school s_i. What is the correct interpretation?

10. Pg. 8, lines 300-301 and Figure 6. The functional form of the points as you move from the k=1 to the k = 10 panel is very regular; basically moving from exponential-like to linear. This suggest to me that there is something fundamental about the geometry of the measurement units that is driving the pattern. It would be very useful to simulate a null model here to know how this measure behaves when there are no correlations in the data. For example, one could assume that all BH events are all 2D Poisson, attacked and non-attacked schools and security installations are also Poisson distributions of points. Then investigate how the measures change as discs of radius k are compared.

11. Pg. 13, lines 463-466. Finding 5. It seems unlikely for such extreme events, but is it possible that there is reporting bias that is correlated with proximity to police stations? That is, attacks farther from formal law enforcement rely on alternative means of solution (such as family, clan groups, local traditional leaders) and therefore are never reported to a formal source (news or police).

12. Pg. 14, lines 494-509 (and Table 4 on pg. 13). It is hard to intuit the rank order arrangement of distance to nth-closest security installation. It feels like there is some interaction between the closeness of security installations and the area size implied by k, but there’s no rhyme nor reason to it. Sure, it is ok to basically say this variable has predictive power, but it would also be nice to have a better understanding why in a behavioral (and geographic) sense certain orderings matter as one scale and not another. For example, why would the 3rd, 4th and 5th closest be important at k = 5, but not 1 and 2. Some explanation seems needed.

13. Pp. 13-14. The statistical testing associated with the decision tree analyses could use better explanation.

Reviewer #6: Overall, this is a very will written manuscript, doing a great job at spatially analyzing Boko Haram's attacks on Schools. There are only two concerns that I have in improving the manuscript.

First, the argument for hypothesis #3 is quite weak and the author(s) need to bolster the rationale for it.

The other addition that I would suggest is for the author(s) explicitly state what the unit of analysis is (i.e. the areal unit). Is it the school? The ward?

6. PLOS authors have the option to publish the peer review history of their article (what does this mean?). If published, this will include your full peer review and any attached files.

Reviewer #1: No

Reviewer #2: **Yes: **Linchen He

Reviewer #3: No

Reviewer #4: No

Reviewer #5: No

Reviewer #6: No

---

## [Author Response · Author response to Decision Letter 1]

5 Feb 2025

We would like to thank all the reviewers for their insightful comments. We found the comments extremely helpful in improving the framing of the main results, the analysis, presentation of the paper, and embedding it within the context of related work in criminology and sociology. Thanks so much! The Comment Number (X.Y) indicates that we are addressing the Y’th comment made by reviewer #X. Thanks again – and we hope these changes address your concerns!

REFEREE #1:

1.0. Thanks for your kind words.

1.1. Thank you for flagging this. It has been fixed.

1.2. Thanks for your kind words. We have included a paragraph on this in the Conclusion. Thanks for the suggestion.

REFEREE #2:

2.0. Thanks for the kind words.

2.1. Thank you for pointing this out! We have now fixed it.

2.2. Thank you for this comment. These “Risk Scores” were not created by us but were provided by the GRID3 Hub. We have updated the Socio-Economic data section to provide some more information about it. Thanks!

2.3. Figures 6 and 8 are in the correct order. We have updated the figures for better resolution. Thanks for the helpful comment.

2.4. Thank you for pointing this out. Table 1B summarizes urban schools. The typo has been fixed in the manuscript.

2.5. We should have explained it better in the initial version of the paper. All the ML-based analysis involves a standard 10-fold cross validation analysis with training/validation done on 9 folds and testing on the 10th holdout fold. So, there shouldn’t be any overfitting. We have added a few sentences on this in the “Predictive Performance” paragraph of the “Machine Learning Based Analysis” section. We are glad you asked about this as we should have made this clear early on. Please also see the paragraph in the new “Limitations” section (just before the Conclusion”. Our prior book on Boko Haram shows prediction on WHEN different types of attacks will occur. This work focuses specifically on WHERE. Predicting both WHEN and WHERE at the same time is too difficult because of the paucity of data – 76 school attacks over 14 years (so about 5.4 school attacks per year on average).

2.6. Fixed, thank you.

2.7. We have added a paragraph at the beginning of the “Multivariate Machine Learning Inspired Statistical Inferences of Boko Haram School Attacks” to make this connection. We have also significantly expanded the third paragraph of that section. Thanks for the suggestion.

2.8. Thanks for the excellent suggestion. We have now computed Positive Predictive Values (PPV), Negative Predictive Values (NPV), Sensitivity, and Specificity as additional evaluation metrics for the relevant analyses. We believe these additional metrics, as the reviewer kindly suggested, may add additional insights about our machine learning models and statistical inference. Thus, we now include it as Supplemental Table 14 and 15 in Appendix G.

2.9. The risks shown in Figures 1 and 2 are based on the probabilities returned by the best performing ML model, namely AdaBoost. We have updated the captions of Figures 1 and 2 to reflect this. Thanks for the suggestion.

REFEREE #3:

3.0. Thanks so much for your kind and encouraging words.

3.1. Thank you for your comment. A time-series analysis was performed on the time-series version of the dataset using LSTM-based models. However, due to the huge imbalance in the dataset (there were only 76 out of a total of over 103,000 schools were attacked), the LSTM fails to capture the temporal dynamics well, by overfitting in favor of the “No Conflicts” class. For more details on the performance metrics, please refer to Appendix H.

3.2. Thank you for your comment, it has been fixed in the manuscript.

3.3. We have added a paragraph at the beginning of the section on “Machine Learning-based Analysis”. Thanks for the suggestion!

3.4. Please note that all the features selected are based on theory from the social science literature. We have added a paragraph at the beginning of the section on “Machine Learning-based Analysis”. Thanks!

3.5. We wanted to investigate how predictions of whether a school is at risk of attack vary in quality as we expand the window of what someone might consider threatening to the school. We investigated how predictive performance changes when the threshold is set to 1, 2, 3, 5, 10km ranges. At the 1km threshold, our experiments demonstrate a lack of ability to predict if a school within 1km of a given school S will be attacked. But for the 2km, 3km, 5km, and 10km levels, our best predictive model performs well. We have added a discussion on this toward the beginning of the “Statistical Inference of School Attacks in Relation to Social Attributes” section.

3.6. Thank you for your comment, the plots have been updated with the axis labels.

3.7. Thank you for your comment. The ROC curves’ plot has been added and addressed in page 17 (Figure 14), and you can find the confusion matrices in Appendix H of the supplementary material. For full transparency and reproducibility, when running the AdaBoost model to gather this data, we observed minor variations in precision and recall across different dataset variations (k=1-10). These variations were no more than 2% from the values initially reported and did not impact the originally reported F1-scores. These changes are attributed to updates in the sklearn AdaBoost Classifier module. Consequently, we have specified the version of sklearn used in the paper to ensure clarity and consistency, as well as updated the Table 4 and Appendix B.

3.8. We have added a new section called “LImitations” and addressed this point there. Thanks for the suggestion.

3.9. Thanks. We have added a new Appendix (appendix J) with details on the statistical methods used.

REFEREE #4:

4.1. Thank you for your comment, we have updated the security installations section to address this comment. We have also added more information about the socioeconomic data (but as this data was created by GRID3, not by us, we are limited to publicly available information about it). We have created and included new maps showing the locations of security installations (Figure 6) and socioeconomic characteristics of regions in Nigeria (Figures 7-9).

4.2. Thank you for your comment. We updated the Data section (lines 211-215) and the Socio-Economic Data (288-295) to address it.

4.3. We have added a paragraph at the end of the Conclusions that explains this.

4.4. Thanks so much for the kind words. We appreciate it.

4.5. We have added 2 new paragraphs in a new section called “Limitations”. Thanks for the suggestion.

4.6. Thanks for raising this point. We have added a new paragraph on ethics in the Limitations section.

REFEREE #5:

5.0. Thanks so much for your kind comments and excellent suggestions for improvement.

5.1. Thanks. We have carefully proofread the paper. In addition, we had the paper read by a couple of external people (native English speakers). Hopefully the grammatical errors and typos have now been fixed.

5.2. Thank you for your comment. We have slimmed down the manuscript by moving a portion of our Decision Tree-based statistical analysis to Appendix K . Instead, we are summarizing these hypotheses in the manuscript in a tabular manner (Table 8)

5.3. We have updated page 11 with a new visual (Figure 10) representation of assigning the dependent variables to the schools, along with more clarifications on the process.

5.4. We have added a note to this effect in the second paragraph of the Introduction. Thanks for the great suggestion!

5.5. Thank you for your comment. The manuscript has been updated to incorporate the theory.

5.6. Thank you for your comment. The manuscript has been updated to incorporate the theory.

5.7. Thank you for your comment. The manuscript has been updated to incorporate the theory.

5.8. We have added some rationale for this hypothesis based on the social science literature just before Hypothesis 4. Thanks for pointing it out.

5.9. Thank you for your comment. There were no issues with the data acquisition from ACLED. We also cross-referenced the group’s activity on schools with other online sources. We have now updated the section with more detail (lines 224-226).

5.10. Thank you for this helpful comment. We have extended the Boko Haram section, particularly lines 235-239 to address the issue you have raised.

5.11. The tabular summary has been added on page 6. Thanks for the suggestion!

5.12. Thanks. We are not claiming this is a new measure. It is simply a way of framing an error-measure in terms of a binary classification problem. It is similar to something called distance-based accuracy. We have added a better explanation in two new paragraphs introduced before Hypothesis 1. Thanks for pointing out the lack of clarity.

5.13. We thank reviewer for this comment. In this paper, we treated each radius as independent analysis rather than interdependent entities. Thus, considering all radii together with Poisson methods was not applicable under this approach. Instead, we believe a linear model-based approach would be the more suitable approach as it has greater generalizability. Further, this is also the basis for us to adjust our raw P-values with the Bonferroni method. We hope this is ok!

5.14. This could certainly happen – but we are not aware of any instances where this did happen. We have added a paragraph on this in our new Limitations section. Thanks for the excellent suggestion.

5.15. This is an interesting point, and the exact answer is not clear. We spent some time digging into this, but couldn’t come up with anything conclusive. But regardless of the actual importance rank, all of these features related to proximity to security installations are clearly important. We have added a sentence about the importance of this Table in Finding 5. Thanks.

5.16. We have added Appendices G and J which provides more information on the odds ratio-based testing and also provides additional statistics (beyond odds ratio). Thanks for the suggestion!

REFEREE #6:

6.0. Thanks so much for your very kind words.

6.1. We have now added some supporting rationale before Hypothesis 3. Thanks for the suggestion!

6.2. Thank you for your comment, the clarification was added to the School Data section (line 246).

---

## [Decision Letter · Decision Letter 1]

27 Feb 2025

A Quantitative Geospatial Analysis of the Risk that Boko Haram Will Target a School

PONE-D-24-14163R1

Dear Dr. Subrahmanian,

We’re pleased to inform you that your manuscript has been judged scientifically suitable for publication and will be formally accepted for publication once it meets all outstanding technical requirements.

Kind regards,

Jessica Leight, PhD

Academic Editor

PLOS ONE

Additional Editor Comments (optional):

Reviewers' comments:

Reviewer's Responses to Questions

**Comments to the Author**

1. If the authors have adequately addressed your comments raised in a previous round of review and you feel that this manuscript is now acceptable for publication, you may indicate that here to bypass the “Comments to the Author” section, enter your conflict of interest statement in the “Confidential to Editor” section, and submit your "Accept" recommendation.

Reviewer #2: All comments have been addressed

Reviewer #5: All comments have been addressed

2. Is the manuscript technically sound, and do the data support the conclusions?

Reviewer #2: Partly

Reviewer #5: Yes

3. Has the statistical analysis been performed appropriately and rigorously? 

Reviewer #2: Yes

Reviewer #5: Yes

4. Have the authors made all data underlying the findings in their manuscript fully available?

Reviewer #2: Yes

Reviewer #5: Yes

5. Is the manuscript presented in an intelligible fashion and written in standard English?

Reviewer #2: Yes

Reviewer #5: Yes

6. Review Comments to the Author

Reviewer #2: (No Response)

Reviewer #5: The authors have addressed all of my concerns. I believe this paper is an important contribution to the literature.

7. PLOS authors have the option to publish the peer review history of their article (what does this mean?). If published, this will include your full peer review and any attached files.

Reviewer #2: No

Reviewer #5: No

---

## [Editor Report · Acceptance letter]

PONE-D-24-14163R1

PLOS ONE

Dear Dr. Subrahmanian,

I'm pleased to inform you that your manuscript has been deemed suitable for publication in PLOS ONE. Congratulations! Your manuscript is now being handed over to our production team.

Kind regards,

on behalf of

Dr. Jessica Leight

Academic Editor

PLOS ONE